Journal of Data-centric Machine Learning Research (2024)          Submitted 12/23; Revised 03/24; Published 10/24

# The Matrix Reloaded:
# Towards Counterfactual Group Fairness in Machine Learning

**Mariana Pinto**[*]
*Associação Fraunhofer Portugal Research - AICOS, Porto, Portugal*

**André V. Carreiro** [*]                    ANDRE.CARREIRO@AICOS.FRAUNHOFER.PT
*Associação Fraunhofer Portugal Research - AICOS, Porto, Portugal*

**Pedro Madeira**
*Associação Fraunhofer Portugal Research - AICOS, Porto, Portugal*

**Alberto López**
*INCMLab, Imprensa Nacional - Casa da Moeda, Lisbon, Portugal*
*Mathematics Department and CEMAPRE, ISEG, University of Lisbon, Portugal*

**Hugo Gamboa**
*Associação Fraunhofer Portugal Research - AICOS, Porto, Portugal*
*Laboratory for Instrumentation, Biomedical Eng. and Radiation Physics (LIBPhys-UNL), NOVA School of Science and Technology, Caparica, Portugal*

**Reviewed on OpenReview:** *https: // openreview. net/ forum? id= wqDiPP8Xm7*

**Editor:** Yang Liu

## Abstract

In today's data-driven world, addressing bias is essential to minimize discriminatory outcomes and work toward fairness in machine learning models. This paper presents a novel data-centric framework for bias analysis, harnessing the power of counterfactual reasoning. We detail a process for generating plausible counterfactuals suited for group evaluation, using probabilistic distributions and optionally incorporating domain knowledge, as a more efficient alternative to computationally intensive generative models. Additionally, we introduce the Counterfactual Confusion Matrix, from which we derive a suite of metrics that provide a comprehensive view of a model's behaviour under counterfactual conditions. These metrics offer unique insights into the model's resilience and susceptibility to changes in sensitive attributes, such as sex or race. We demonstrate their utility and complementarity with standard group fairness metrics through experiments on real-world datasets. Our results show that domain knowledge is key, and that our metrics can reveal subtle biases that traditional bias evaluation strategies may overlook, providing a more nuanced understanding of potential model bias.

**Keywords:** Bias, Fairness, Counterfactual, Confusion Matrix, Data Augmentation, Machine Learning

---

[*]. These authors contributed equally to this work.

Table 1: Group Fairness Criteria and Metrics

| Criterion | Fairness Requirement | Metric | Meaning |
|---|---|---|---|
| Independence | Outcome independent of the sensitive feature | DemP | Difference between Predicted Prevalence (Feldman et al. (2015)) |
| Separation | Given an outcome, similar probability of being correct (Equal error rates) | EOdds EOpp PredEq | Error rate diff. for both outcomes Error rate diff. for positive outcomes Error rate diff. for negative outcomes (Hardt et al. (2016)) |
| Sufficiency | Given a prediction, similar probability of being correct | PredP | Difference between Precision (Verma and Rubin (2018)) |

## 1 Introduction

Machine Learning (ML) models have revolutionised decision-making processes across numerous domains. However, these models can mirror or even amplify bias present in the training data, raising concerns about potentially unfair or discriminatory outcomes. Studies often focus on identifying discrimination based on a sensitive feature (e.g., race, sex), also known as a protected attribute in specific applications. Detecting and mitigating bias is critical for equitable and trustworthy ML systems. Nonetheless, there is no consensus on an unequivocal definition of a fair decision in ML, despite numerous philosophical streams emerging over time, including egalitarianism and utilitarianism (Beretta et al. (2019)).

The most prevalent approach for detecting bias relies on group fairness. Existing fairness metrics under this umbrella term are referred to as *parity measures* as they compare performance parameters between the data sectioned by the subgroups of the sensitive feature. These metrics can be further divided by the fairness criterion employed: *Independence*, *Separation*, and *Sufficiency* (Barocas et al. (2023)) (Table 1).

While group fairness targets similar outcomes for different subgroups, another stream evaluates bias in an instance-based approach, defending that similar instances (except for the sensitive feature) should have similar outcomes (Dwork et al. (2011)). This concept of individual fairness is still a disputed methodology. Fleisher (2021) argues the impracticability and insufficiency of the methodology, further defending that the analysis may induce human biases. Still, in this context, we highlight counterfactual fairness, employing the concept of causal inference, as introduced by Kusner et al. (2017). Under this notion, a model is deemed counterfactually fair if, for any sample, the actual prediction is the same as the one generated in a 'Counterfactual world', changing the value of the sensitive feature, while keeping other not causally dependent variables constant.

By definition, counterfactual fairness applies under the premise that bias is found when there is a causal link between a sensitive feature and the outcome. This framework assumes the sensitive feature should not impact the outcome. While this holds true in some cases, it is not universally feasible. As an example, an individual's sex should not influence their college admission decision. On the other hand, sex, or race, might be a biologically relevant factor for clinical trials' enrollment, as well as a source of bias.

## 1.1 Contributions

To broaden the application of counterfactual fairness in diverse scenarios, we introduce a novel framework for bias evaluation that starts by creating plausible counterfactual examples. This approach shifts from strict causal relationships to a more flexible model that changes the sensitive feature while plausibly adjusting (cor)related features. Plausibility is derived both from Probability Distribution Functions (PDFs), and domain knowledge, when available. Rather than enforcing complete independence from the sensitive attribute, our method evaluates independence from characteristics plausibly linked to it. Moreover, recognising the challenges of generating synthetic data and its evaluation, the incorporation of domain knowledge in various forms drives the credibility and relevance of the generated Counterfactuals (CFs).

The proposed framework further comprises a comparison between the model outcomes in both the original and counterfactual scenarios. This analysis is streamlined through a novel Counterfactual Confusion Matrix (CCM) and its extended version. Several metrics, analogous to the ones derived from a traditional confusion matrix, are proposed to aggregate the individual results into group-based insights. This balanced use of statistical distributions and causal insights facilitates a more nuanced transition from individual to group fairness, enriching the robustness and applicability of our framework in different contexts.

## 1.2 Related Work

This subsection highlights the related work of applying counterfactual reasoning to fairness and related applications, focusing on the methodology and challenges involved.

### 1.2.1 COUNTERFACTUAL EXPLANATIONS

We start by discussing the role of Counterfactual Explanations (CFEs) in ML. CFE is a key framework for explainability, utilising "what if" scenarios to improve interpretability. CFs identify minimal feature changes for different outcomes (Wachter et al. (2017)). Despite their widespread use in model-agnostic settings across various data types, standardising evaluation methodologies remains a challenge (Stepin et al. (2021)).

In the context of CFE, there are several studies for generating plausible CFs employing metrics such as *sparsity* and *diversity* (Smyth and Keane (2021)), as well as improving the *consistency* of the generated CF (Black et al. (2021)). Diverse Counterfactual Explanations (DiCE) is another prevalent technique for generating CFs that satisfies two properties: diversity and feasibility given the user context and constraints (Mothilal et al. (2020)). The method is able to generate a diverse set of CFs that effectively approximate local decision boundaries providing meaningful insights into the model's predictions. Human-in-the-Loop processes also contribute to robustness by incorporating domain expertise (Kaushik et al. (2019)). Despite these advancements, a gap persists in applying CFE methodologies to counterfactual fairness due to conflicting goals. Most CFE generation models aim to flip the prediction, while counterfactual fairness focuses on changing the sensitive feature and observing prediction changes.

### 1.2.2 COUNTERFACTUAL FEATURES VERSUS OUTCOMES

There are two main approaches to achieving counterfactual fairness. The first, more common approach previously introduced, considers CFs of sensitive features, while the second focuses on counterfactual outcomes. The former, with which this work aligns, involves changing a sensitive feature (e.g., sex or race) and observe whether the model's distribution is changed (Kusner et al. (2018); Cornacchia et al. (2023a,b); Russell et al. (2017a)). In contrast, the latter involves estimating a change in the outcome and observing the effect on the predictor's distribution. This approach is often studied in Risk Assessment Instruments (Coston et al. (2020, 2021); Mishler et al. (2021); Mishler and Kennedy (2021)).

However, as mentioned before, the challenge of developing accurate causal models, especially with confounding factors, remains significant (Russell et al. (2017a); Kusner et al. (2018); Cornacchia et al. (2023a,b)).

### 1.2.3 COUNTERFACTUAL BIAS EVALUATION

Regarding bias evaluation, Coston et al. (2020) propose counterfactual analogues of common performance and fairness metrics, introducing doubly robust estimation for calculating them. They use counterfactual outcomes, which would have been observed under a different decision policy, and a treatment/decision model that predicts the treatment or decision based on the features.

Mishler et al. (2021) propose a post-processing method to achieve counterfactual equalized odds in Risk Assessment Instruments, also computed using doubly robust estimators. These last approaches fall under the category of counterfactual outcomes, while our proposed framework focuses on counterfactual (sensitive) features.

Cornacchia et al. (2023a) propose a counterfactual generation tool to study implicit bias in predictive models even when sensitive features are removed. Their approach allows to identify proxy features, defining a metric called Counterfactual Flips, representing the percentage of the generated CFs that belong to different demographic groups through a sensitive feature predictor. The research group further presents a novel set of fairness metrics including the Counterfactual Fair Opportunity (CFO), a Discounted Cumulative Counterfactual Fairness, and its normalised version (Cornacchia et al. (2023b)).

Maughan et al. (2022) devised a method to evaluate CF fairness after deployment, suited for Neural Networks (NN). The proposed metric, Predictive Sensitivity, measures the dot product of the gradient of the feature's contributions in the task classifier and in predicting the protected attribute (auxiliary model). Both factors are calculated by the difference in weight of the feature when flipping the protected attribute. This measure is proposed as a proxy to causal inference and higher values indicate CF bias. During training, a base acceptable value for predictive sensitivity is established, and, after deployment, predictions are monitored to detect if the model is striving away from the acceptable range. This method does not require an explicit CF generation process, being a cost-efficient approach. Nonetheless, it requires an additional model and white box access for auditing and is limited to NNs.

Despite these developments, there remains a gap in comprehensive metrics for conducting an in-depth global analysis of decision flips in counterfactual fairness applications. The work of Black et al. (2020) presents a notable contribution in this regard, introducing

'fliptests' that account for flips from positive to negative or vice versa for each subgroup, further constraining by features as well as ground truth based on the case study. However, this evaluation focuses on understanding which features changed in the CF generation process instead of measuring model prediction's flips. The generation process uses Optimal Transport Maps (OTMs) or approximations through Generative Adversarial Networks (GANs) to convert distributions between subgroups, effectively achieving feasible CFs. Still, these methods have shortcomings: they are computationally demanding and, while effective in reading and learning the distributions, they are easily prone to replicating statistical relations that are not necessarily desirable or even representative of real-world populations. This is used to their advantage by identifying the features that change the most for a given *fliptest* in order to understand the underlying tendencies of the model. Goethals et al. (2023) propose another metric based on CF flips, the *PreCoF* which measures the difference in portion of these flips.

Albeit with similar methods and goals, our work aims for a more straightforward evaluation based on plausible CFs. Instead of focusing on causal links and ensuring that the sensitive feature does not affect the model prediction, our aim aligns with the fundamental idea behind counterfactual thinking. While ambitious, we strive to provide an evaluation framework based on CFs that mirrors the original dataset in a 'Counterfactual World'. With this, each CF should only stray away from the original sample in terms of group membership and characteristics we can argue would be different. To achieve this, we propose a controlled CF generation process, described in Section 2.1. Our goal is to set new necessary, albeit not sufficient, conditions to define a model as unbiased.

### 1.2.4 COUNTERFACTUAL BIAS MITIGATION

Regarding bias mitigation techniques employing CFs, Russell et al. (2017b) proposed a method to make fair predictions by integrating multiple causal models. In that study, the model is generated with an optimisation task of achieving, as the authors define it, an *Approximation Counterfactual Fairness*. This metric is calculated based on the difference of flipped instances between each sequential causal model.

Concerning data augmentation, several studies explored the benefits of utilising CFs for attenuating the class imbalance (Pombo et al. (2023); Temraz and Keane (2022)), as well as testing which generation processes ensure better results (Kaushik et al. (2020)), which can be expected to translate well into minimising counterfactual bias. Our proposed framework also investigates the impact of counterfactual augmentation in bias mitigation, guided by the proposed metrics and domain knowledge.

## 2 Methodology

### 2.1 Generating Plausible Counterfactual Fairness

The goal of this process is to generate, for each sample, corresponding CFs (relative to a sensitive feature) that share the same outcome. Additionally, features highly correlated with the sensitive feature are adjusted to better fit the counterfactual subgroup. The proposed algorithm considers the statistical distribution of each sensitive subgroup relative to each class to adjust feature values.

In scale, for a given sensitive attribute with $G$ possible subgroups, at least one CF is generated for each different subgroup for each sample, resulting in at least $G - 1$ CFs per sample.

To introduce the method, we characterize the dataset by a set of features $F$, a sensitive attribute with multiple subgroups $g \in G$, and the target variable with a set of possible labels $Y$ (for a classification task). We define the group of samples from which we want to generate CFs as $S$ and the training set, used as a reference in the generation process, as $S_{train}$. For ease of interpretation, the proposed process is described in the context of a binary classification problem, $y \in \{0, 1\}$, where 0 corresponds to the negative (-) class and 1 to the positive (+) class, and a binary sensitive feature with subgroups $g \in \{A, B\}$. However, the method generalizes to multiclass problems and categorical sensitive features, as discussed in Appendix C.

The algorithm starts by handling continuous variables. The process (considering only one label) is roughly illustrated in Figure 1. The generation begins by computing the Cumulative Distribution Functions (CDFs) $P \in \{P_{A-}, P_{A+}, P_{B-}, P_{B+}\}$ for the values of each continuous feature $f \in F$ of the training set $S_{train}$. For each sample $s_k$ in the sample space $S$ belonging to subgroup $A$ that has value $v_{k,i}$ for a continuous feature $f_i$ and label $y_k = 1$, the goal is to generate a CF in the subgroup $B$ with a modified value $v'_{k,i}$, retaining the label $y_k = 1$. The algorithm assesses the probability of the event $x \leq x_n$ within the $P_{A+}$ distribution (CDF of training samples from subgroup $A$ with positive label) (Figure 1B). Defining $x$ as the sorted possible values for the feature $f_i$, where $v_{k,i}$ corresponds to its $n$th value, $x_n = v_{k,i}$. In case $x_n$ is not represented in the distribution, the method uses interpolation to infer the approximate probability. Defining the closest values in $P_{A+}$ as $x_{n-1}$ and $x_{n+1}$ where $x_{n-1} < x_n < x_{n+1}$, formally, for the sample $s_k \in S_{A+}$, the probability of event $v_{k,i} = x_n$ is given by Equation 1.

$$P_{A+}(v_{k,i}) = P_{A+}(x_n) = P_{A+}(x_{n-1}) + (P_{A+}(x_{n+1}) - P_{A+}(x_{n-1}))\frac{x_n - x_{n-1}}{x_{n+1} - x_{n-1}} \quad (1)$$

Then, it searches for the corresponding value $v'_{k,i} = x_m$ in the distribution $P_{B+}$ from $P_{A+}(x_n)$ (Figure 1C). When not represented, the value is inferred by interpolation, formally defined in Equation 2.

$$v'_{k,i} = x_m = x_{m-1} + (x_{m+1} - x_{m-1})\frac{P_{A+}(x_n) - P_{B+}(x_{m-1})}{P_{B+}(x_{m+1}) - P_{B+}(x_{m-1})} \quad (2)$$

A similar approach is adopted for categorical features. However, when finding the corresponding feature values of the CF, instead of interpolation, the method selects the nearest value from the new distribution, considering the discrete nature of the feature. It is important to note that this process is specifically suited for features - whether continuous or categorical - that exhibit an ordinal relationship, such as 'age' or 'burn degree'. For other categorical features, it is preferable to one-hot encode them and treat them as binary. Importantly, this method is able to treat encoded features as a group, ensuring the absence of impossible combinations.

To provide a clearer picture with an example, consider the task of generating a CF of a male patient based on a healthy female patient's data. It would be implausible if the

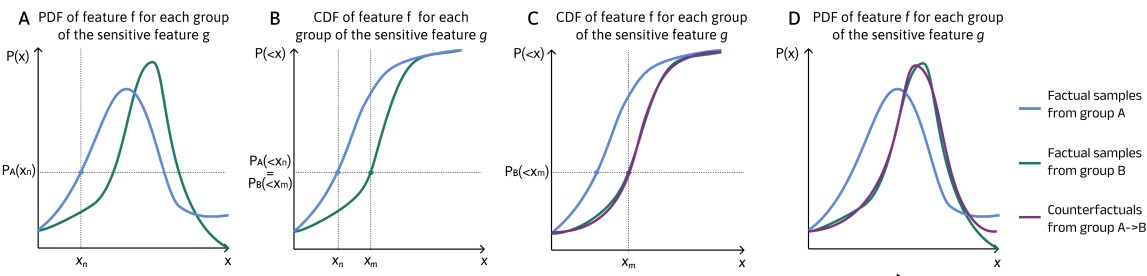

Figure 1: Sequential steps for adjusting the value $x_n$ of the continuous feature $f$ throughout the generation of CF from group $B$ from samples from group $A$, given the sensitive feature $g \in A, B$. Note that the process further sections the samples by label. A - Estimate PDF of the continuous feature sectioned by subgroups and label; B - Estimate CDF for both distributions marking the corresponding value $x_m$ in the other group CDF; C - Resulting CF CDF(purple) when repeating the process for every sample; D - Resulting PDF from the CFs(purple).

testosterone levels in the CF remained the same as the original female patient's levels. The proposed method finds the testosterone concentration of the initial female sample within its distribution, say, within the higher quartile, and grants that the male CF is within the corresponding quartile. So, a female with comparatively higher testosterone levels corresponds to a male with relatively higher levels, and the converse holds.

The process of flipping binary features is based on the probability of the original value, conditioned for each subgroup. Initially, for each binary feature, the method tests the probability of the original feature value $v_i$ occurring in the new group. Should this probability fall beneath a predetermined threshold, then the feature value is flipped. This step prevents the emergence of 'impossible' feature combinations. For example, consider an original sample representing a pregnant woman, and we aim to generate a male CF. Since the attribute of pregnancy is female-exclusive, the probability of a pregnant individual being female is 1, while the counterpart for a male is 0. Consequently, the algorithm modifies the feature value from 'pregnant' to 'not pregnant' in the male CF.

Subsequently, an iterative test for the remaining binary features initiates. For a sample $s_k$ labelled as 1 from subgroup $A$ with value $v_{k,i}$ for the feature $f_i \in F_B$, where $F_B$ is the set of binary features, the method calculates the difference between the probability of that event occurring for subgroup $A$ and subgroup $B$. If the difference surpasses a predefined threshold ($\tau = 0.5$ by default), the feature value flips, remaining unchanged otherwise. In notation,

$$v'_{k,i} = \begin{cases} |v_{k,i} - 1| & \text{if} |P(v_{k,i}|g = A, y = 1) - P(v_{k,i}|g = B, y = 1)| \geq \tau \\ v_{k,i} & \text{if} |P(v_{k,i}|g = A, y = 1) - P(v_{k,i}|g = B, y = 1)| < \tau \end{cases} \tag{3}$$

Acknowledging that this change might not be plausible, the procedure iteratively refines itself, incorporating additional constraints on subsequent feature flips. In the second level, where $f_i$ flipped, other features are tested, comparing probabilities conditioned on the new

feature value and the sensitive feature. For instance, for the next feature $f_j$, the value $v'_{k,j}$ shall be given in relation to $v'_{k,i}$, following:

$$v'_{k,j} = \begin{cases} |v_{k,j} - 1| & \text{if} |P(v_{k,j}|g = B, f_i = v'_{k,i}, y = 1) - P(v_{k,j}|g = B, f_i = |v'_{k,i} - 1|, y = 1)| \geq \tau \\ v_{k,j} & \text{if} |P(v_{k,j}|g = B, f_i = v'_{k,i}, y = 1) - P(v_{k,j}|g = B, f_i = |v'_{k,i} - 1|, y = 1)| < \tau \end{cases}$$
(4)

As the iterations progress, the pool of samples adhering to these constraints diminishes. Consequently, the probabilities become less representative of the population. Thus, it is crucial to consider the size of the dataset and select an appropriate *depth* - a hyperparameter that determines the maximum number of iterations. Because there are features that can be established as unrelated to the sensitive feature, it is possible to select the potential features and keep the others unchanged.

This approach can yield multiple CFs for each sample as it evaluates each feature in parallel, depending on the hyperparameters. To filter out inconsistent CFs, a supplementary optional validation phase is introduced. This phase examines the feature distributions and discards the less likely candidates. First, Principal Component Analysis (PCA) is applied to the dataset, which is then sectioned by label and group membership. For each possible combination of $n - 1$ principal components, the centroid is calculated (effectively projecting the remaining components onto a single one). CFs are marked as invalid if they are farther from one of the centroids than a set threshold, based on a number of standard deviations (e.g., $1.5 \times \sigma$). The generated CFs can then be used to investigate the presence of counterfactual bias in a ML model by comparing its outcomes for both the original samples (e.g. a validation set) versus their counterfactual versions. The main components of this methodology are comprised in Algorithm 1, found in Appendix A.

## 2.2 Counterfactual World Building Considerations

This methodology requires a clear idealisation of a counterfactual world and the changes it may entail. Deciding which features are rightly related to the sensitive feature can be challenging and sometimes inconclusive. For instance, there may be characteristics that are rightly related to a group in one scenario but not in another. In these circumstances, valuable insights can still be gleaned from counterfactuals, namely through naive and greedy approaches. A naive approach would involve only flipping the value of the sensitive feature, while the greedy approach would allow changes in all features related to the group. While plausibility is not guaranteed, the proposed metrics can still provide hints of potential biases. The naive approach will display an approximation of the direct importance of the sensitive feature in the dataset. In contrast, the greedy approach refers to the global impact of the represented group's distributions on the decision. In some instances, the latter may suffice for the purposes of the project. For example, in Section 3.1, we present a scenario where the counterfactual world is not clear-cut since the correlations with the sensitive feature are not direct. To surpass the issue, we start by identifying the few relations we are sure do not have a plausible effect, then attempt to decorrelate them (creating a new base dataset), followed by a greedy approach, and finally empirically choose the counterfactuals we are the most confident. Different methods may be employed to validate counterfactuals; a safe approach is to adapt synthetic data validation metrics so that while

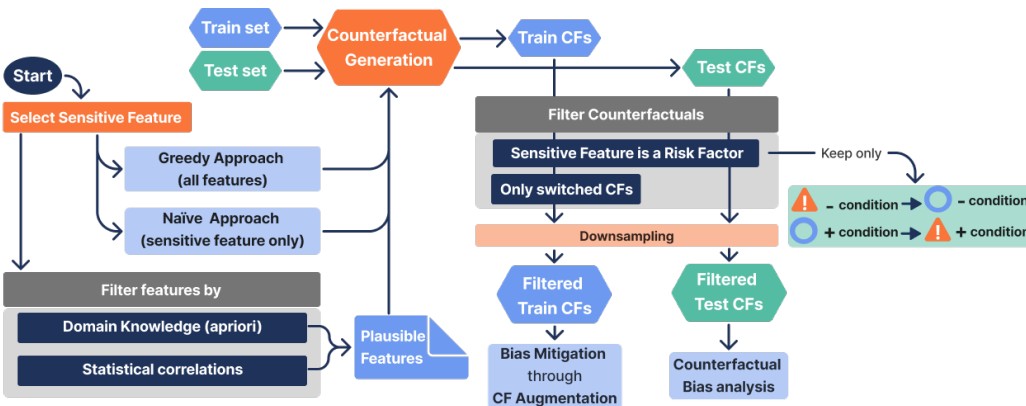

Figure 2: Flowchart of the proposed CFs Generation process, characterising the feature selection process, as well as the CFs filtering options.

they may not be definite plausible mirror images of the original samples, they can at least be assumed to be feasible instances. Additionally, since the proposed metrics are intended to complement other fairness methods, such as group fairness, assessing the effectiveness of data augmentation with more objective metrics, namely group fairness parities, is also possible. We include a flowchart with general interpretations to guide the reader through the preprocessing steps to filter plausible features, such as choosing the CFs for evaluation and augmentation (Figure 2).

### 2.3 The Counterfactual Confusion Matrix

The prominent Confusion Matrix (CM) allows for easy visualisation of the combinations of the ground truth and the predictions, containing all the necessary information for evaluating the performance of a classification problem. Inspired by this instrument, the proposed CCM adds two dimensions to the analysis: the sensitive feature values and the resulting prediction for the counterfactual samples. As follows, to simplify interpretation in specific applications, we propose two versions of the matrix: a simplified format that omits the ground truth, and an extended version.

By design, the assumption of independence in generating the CFs implies that, for a given sample $s \in S$, the label for the pair $s, s_{CF}$ remains the same  Thus, it is possible to directly compare each pair with a criterion of **consistency**, evaluating if the prediction changes for the CF sample. There are four possible combinations of results for a binary problem, which can be counted and presented just like the combination between predictions versus real values can be summarised in a CM. In this case, the reference is the original prediction, so it occupies the spot of the ground truth in rows, while the CF predictions are placed in the columns, as illustrated in Figure 3.

When analysing a counterfactual matrix, the base parameters account for the CFs predicted as positive (P) or negative (N) which are consistent (C) with the original sample

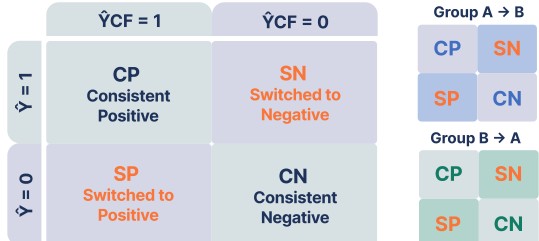

Figure 3: The Counterfactual Confusion Matrix (CCM) with two additional CCMs represented(right) highlighting the importance of drawing these matrices by group membership.
$\hat{Y}$ - Prediction for the factual(original) sample; $\hat{Y}_{CF}$ - Prediction for the CF sample.

or switch (S) its value following the nomenclature: Consistent Positive (CP), Consistent Negative (CN), Switched to Positive (SP), and Switched to Negative (SN).

### 2.3.1 Derived Metrics

Several metrics can be derived from the CCM, some in direct analogy to metrics computed from the traditional CM. We will define the proposed metrics, provide the analogue in the traditional CM, if it exists, and describe its meaning when applied to bias analysis. When appropriate, we further define the complement metric ($comp\_metric = 1 - metric$).

- **Consistency Rate (CR)**: The analogue to Accuracy (ACC) in the CM, this measures the proportion of instances where the original prediction remained the same in the counterfactual scenario. It is calculated as $(CP + CN)/(CP + CN + SP + SN)$. Its complement is the **Switch Rate (SR)**, calculated as $(SP + SN)/(CP + CN + SP + SN)$. Although measuring the consistency of counterfactual predictions is not new (Cornacchia et al. (2023a)), we formalise it based on the proposed CCM.

- **Positive Switch Rate (PSR)**: Measures the proportion of instances originally predicted as negative that switched to a positive outcome in the counterfactual scenario. It is calculated as $SP/(SP + CN)$ and stands as the equivalent to the False Positive Rate (FPR) in the CM. Its complement can be defined as **Negative Consistency Rate (NCR)**, whose equivalent in the CM is the Specificity.

- **Negative Switch Rate (NSR)**: Inversely, the NSR captures the fraction of original positive predictions that switch to negative after the counterfactual changes. It is calculated as $SN/(SN + CP)$. The analogue in the CM is the False Negative Rate (FNR). Its complement is defined as **Positive Consistency Rate (PCR)** whose CM analogue is the Sensitivity or Recall.

- **Positive Consistent Precision (PCP)**: This metric is equivalent to the Precision in the CM, and is computed as $CP/(CP+SP)$. It can be interpreted as the proportion

of positive counterfactual predictions that are consistent before and after the counterfactual changes. Its complement is defined as the **Positive Switch Discovery Rate (PSDR)**, whose analog in the CM is the False Discovery Rate (FDR).

- **Counterfactual Matthew's Correlation Coefficient (CMCC)**: The counterfactual version of the Matthew's Correlation Coefficient (MCC) measures the alignment between the original and counterfactual predictions. It keeps the favourable properties of the original MCC, like being robust to unbalanced data (in this case the predictions, not the ground truth). It can be computed as $\frac{CP \times CN - SP \times SN}{\sqrt{(CP+SP) \times (CP+SN) \times (CN+SP) \times (CN+SN)}}$.

These metrics provide a comprehensive view of the model's performance under counterfactual conditions. To better assess the potential bias, one should compute these metrics for the different subgroups of the sensitive attribute (e.g., males and females when *Sex* is the attribute) and compare the obtained results. One could use the absolute difference to get a measure of group disparity or use a ratio for a relative disparity, especially if higher sensitivity to small changes is desired.

For some of the metrics, higher values for one of the subgroups may suggest that the model is biased against that subgroup, like the PSR, and PCP. For metrics like the NSR, higher values for a subgroup may indicate that the predictions are biased in favour of that subgroup. High values of the SR may indicate the presence of bias, but it's difficult to assess in which direction. Higher values for the CR, NCR, PCR, and PCP align with a counterfactually fair model.

However, these metrics do not account for the ground truth. This allows to study model bias independent from knowing the real labels of the population, especially important when a system is in production. On the other hand, it may miss important context such as the real prevalence or model correctness in specific subgroups in the training set, to better evaluate if the model minimises, maintains, or amplifies bias.

### 2.3.2 The Extended Counterfactual Confusion Matrix

We propose an Extended Counterfactual Confusion Matrix (ECCM) to allow the visualisation of the relationship between the ground truth labels, or actual outcome prevalence, with both the original and counterfactual predictions. In the binary scenario, the ECCM is a $2 \times 4$ matrix, defined in Figure 4. Its structure expands the CM, dividing each of the original matrix cells into two (consistent and switched). For instance, the first two cells of the first row (True Consistent Positives (TCP) and True Switched Negatives (TSN)) sum up to the known True Positive (TP), and the two cells below (False Consistent Positives (FCP) and False Switched Negatives (FSN)) the False Positive (FP). In turn, we note that summing each 2-cell column results in the four cells of the CCM: CP, SN, SP, and CN.

Since the ECCM purpose is to easily evaluate CF bias, it is better utilised when drawn for each group of the sensitive feature. For this reason, we propose a version of the ECCM broken down into groups, depicted in Figure 11 in Appendix B. The inclusion of the real outcomes allows to understand which instances are more challenging for the model. For instance, if the switches occur mostly on initially correctly predicted samples, it indicates the model is not well adapted to the new (counterfactual) subgroup.

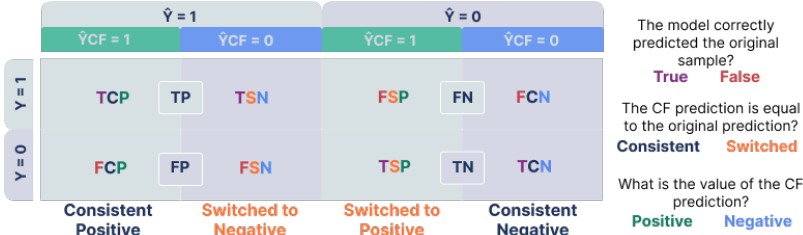

Figure 4: The Extended Counterfactual Confusion Matrix (ECCM).
$Y$ - Ground Truth; $\hat{Y}$ - Prediction for the factual(original) sample; $\hat{Y}_{CF}$ - Prediction for the CF sample.

Considering the additional information about the actual outcomes, a new set of metrics can be defined:

- **True Switch Negative Rate (TSNR)**: Calculated as $TSN/(TSN + FSN)$, it measures the proportion of instances originally correctly predicted as positives whose CF predictions switch to negative. The complement is the **False Switch Negative Rate (FSNR)**. Analogously, we can define them for SP: the **True Switch Positive Rate (TSPR)** and the **False Switch Positive Rate (FSPR)**.

  These metrics allow a more detailed analysis on whether the counterfactual switches are more prevalent in originally correct or wrong predictions (e.g., SN can originate from TP or FP). However, we argue they are insufficient when studied by themselves.

- **True Positive Switch Rate (TPSR)**: This measures the proportion of TP that end up switching (to negative) in the counterfactual setting. It's computed as $TSN/TP$. Similarly, we can define **False Positive Switch Rate (FPSR)** as the proportion of FP that switch to negative in the counterfactual scenario, as computed by $FSN/FP$. Additionally, we can define the equivalent metrics for the negative predictions: the **True Negative Switch Rate (TNSR)** and the **False Negative Switch Rate (FNSR)**.

By comparing their values within a specific subgroup or between different subgroups, we can get a more comprehensive understanding of whether the model is more biased when returning correct or incorrect predictions. This could help focus the mitigation efforts on the instances where the model currently fails, by collecting more diverse training data in these samples' neighborhoods, adjusting the model's parameters, or other targeted bias mitigation techniques. A summary of the proposed metrics can be found in Table 7 in Appendix C.3.

Given the extensive suite of metrics and derived interpretations, we include a flowchart of the general evaluation process using our metrics, starting with those that evaluate the overall consistency followed by more targeted metrics that highlight tendencies (Figure 5). As a general approach, the evaluation starts within a subgroup by evaluating the overall flips that occurred, given either by CR or CMCC. Then, for each class, one may verify if there is any noticeable tendency with PSR and NSR. If the data is not labelled or if we

suspect label biases, the evaluation might end here. Otherwise, we can draw the extended version of the CCM to have a finer look into the switch patterns. As a rule of thumb, switches in TP or True Negative (TN), marked by TPSR and TNSR, are more telling as the model was confidently correct in these instances, and the flips are more alarming. However, complementing with FPSR and FNSR may help identify the best course of action and if the problem is more an issue of optimisation or lack of representation.

To make the suggested workflow more specific, let's consider a hypothetical scenario, of a loan approval model.

**Step 1: Problem Definition**

**Context:** The loan approval model was trained on a dataset containing features like income, credit score, and a sensitive attribute such as race.

**Goal:** Ensure eligible individuals are granted a loan with minimal occurrences of ineligible individuals that receive loans (FPs). Thus, we wish to optimize the precision.

**Fairness concern:** Minimising differences in precision between sensitive groups (PredP).

**Step 2: Model Evaluation Using the ECCM**

**Action:** Apply the ECCM to compare the model's predictions with CF predictions where the race attribute is altered and other features are adjusted to remain plausible.

**Group Fairness:** Let's assume the model has minimal bias in terms of PredP.

**Counterfactual Metrics:** The ECCM reveals that individuals from a particular racial group are consistently less likely to receive loan approval in counterfactual scenarios (e.g., high NSR) for this group). Other metrics can suggest that the change in group membership made the model deny the loan to several eligible individuals (high TPSR for this group) and wrongly attribute more loans to ineligible individuals of the other group (high TNSR for the latter group).

**Interpretation:** General bias against a group where the model 'favours' the other.

**Step 3: Bias Mitigation**

**Action:** Since group fairness metrics did not reveal significant bias, mitigate counterfactual metrics using data augmentation with CFs.

**Step 4: Iteration**

**Action:** CF-based augmentation may have an impact on group fairness metrics. Go back to Step 2 and reevaluate accordingly.

**Possible Problem:** Increase in group bias and decrease in performance. In this scenario, a possible solution would be to reduce the set of CFs used for augmentation, using a selection technique appropriate to the problem objective. In this case, selecting a small number of CF samples for which the model switches the output from TPs to False Negatives (FNs) may be sufficient to achieve better results.

### 2.4 Probability-based Counterfactual Analysis

In classification tasks, the model output typically is a score representing the model's confidence or an uncalibrated probability. To reach a final outcome, required for the previous analyses, a threshold is applied (50% by default in binary problems). However, even when the original and counterfactual predicted outcomes are identical (CR = 100%), important insights may be overlooked. As an example, and assuming the usual decision threshold of 50%, if positive samples in a subgroup score 90%, but drop to 55% in the counterfactual

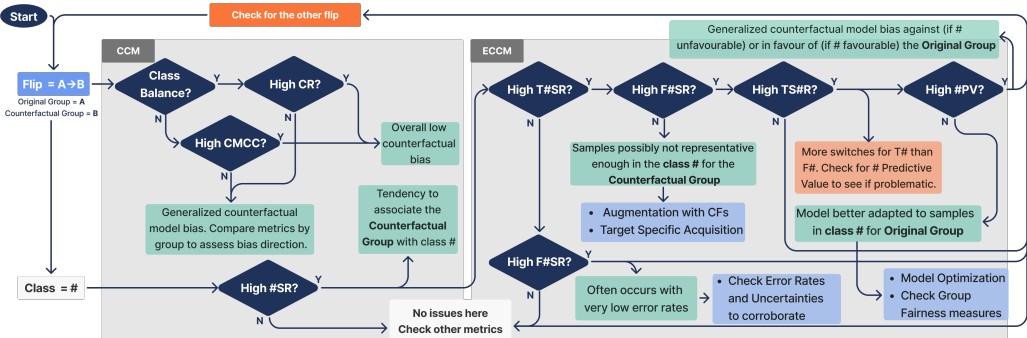

Figure 5: Flowchart of the evaluation process using the proposed Counterfactual Metrics, abiding by a general approach. Green boxes offer some guides on interpreting results, whereas blue boxes suggest some actions to perform based on the results.

setting, the binary decision remains consistent, yet the variation in scores indicates potential bias. To address this, we introduce new metrics that consider these underlying probability scores.

### 2.4.1 MEAN COUNTERFACTUAL DIFFERENCES

Let $m(s_k)$ be the output of model $m$ for sample $s_k$, or its confidence score. Additionally, $s_k^{CF}$ is the counterfactual version of sample $s_k$, where the sensitive feature is changed, and correlated features adjusted for plausibility. We can define Root Mean Squared Counterfactual Differences (RMSCD) that is ananalogueg to the Root Mean Squared Error (RMSE) used in error analysis:

$$\sqrt{\frac{1}{n}\sum_{k=1}^{n}(m(s_k^{CF}) - m(s_k))^2} \tag{5}$$

We chose specifically the RMSE version since in this case we're dealing with values in the interval $[0, 1]$. In this context, the differences, in percentage, are more easily interpreted when using the root mean squared differences.

### 2.4.2 DISTRIBUTION SHIFTS

Another approach to compare confidence scores' differences in the original versus the counterfactual scenarios is to analyse how the score distributions deviate. In case the outcome is entirely independent of the sensitive feature, both distributions should be identical (minus some possibly negligible random factor). One way to achieve this goal is to use the Kullback–Leibler Divergence (KLD), also known as relative entropy and defined as $D_{KL}(P||Q)$, to measure how a PDF $P$ deviates from a second distribution $Q$.

$$D_{KL}(P||Q) = \sum_{i=1}^{n} P(i) \log \frac{P(i)}{Q(i)} \tag{6}$$

In our counterfactual context, we can define $P$ as the distribution of original prediction scores, and $Q$ the corresponding distribution for the counterfactual predictions. $D_{KL}$ is not a metric, in the sense that it is not symmetric, and we choose this ordering, since usually $P$ represents the observations, while $Q$ might represent a theory or a model of $P$. As expected, a high value for $D_{KL}$ as computed for a specific subgroup suggests the presence of model bias. If a proper (symmetric) metric is desirable, we can take the average of $D_{KL}(P||M)$ and $D_{KL}(Q||M)$, where $M = \frac{1}{2}(P+Q)$ (the average distribution), known as the Jensen-Shannon Divergence (JSD), introducing the Jensen-Shannon Counterfactual Divergence (JSCD).

## 2.5 Use Cases

While the goal is to analyse model bias, the proposed framework is model-agnostic and data-centric. As such, this subsection details the datasets employed for testing, outlining the objectives and key aspects for a comprehensive evaluation.

To ensure robust metrics and fair evaluation across different model architectures, we employed a cross-validation approach. Each dataset was divided into five train-test folds, with each fold used for testing exactly once, resulting in five distinct models trained on the remaining folds. Since there is no overlap between test samples, the results from individual test folds were aggregated into a single matrix, allowing for more reliable conclusions due to the increased sample size.

We then describe two chosen use cases, both with clinical applications: one using private data and the other using a publicly available dataset, highlighting the different facets of our framework.

### 2.5.1 CARDIOFOLLOW.AI

*CardioFollow.AI* is a project aimed at providing continuous remote care to patients following hospital discharge after cardiac surgery (Ribeiro et al. (2023); Santos et al. (2023); Curioso et al. (2023)). The dataset includes a broad range of information, such as demographic data, preoperative risks, pre-existing conditions, procedural details, and chronological data (e.g., pre- and post-operative periods, time in intensive care, and time until discharge). The target variable is the occurrence of post-surgery complications, with a significant class imbalance (7.4% positives, 92.6% negatives). Further details can be found in Appendix D.

The current model in practice was optimised for prompt positive outcome identification, using a minimal eight-feature set. It relies on a Random Forest classifier, trained with Threshold Optimisation with 5-fold cross-validation to maximise recall, which is crucial for timely preventive healthcare interventions. The following experiments are based on this model's architecture.

### 2.5.2 HEART DISEASE

*Heart Disease* is a public dataset from the UCI Machine Learning Repository (Asuncion and Newman (2007)). It contains clinical and non-invasive test records from 303 patients

at the Cleveland Clinic, 425 patients at the Hungarian Institute of Cardiology, 200 patients at the Veterans Administration Medical Center in Long Beach, California, and 143 patients from the University Hospitals in Zurich and Basel, Switzerland (Detrano et al. (1989)). The goal is to predict Coronary Artery Disease (CAD) attested by angiography results, graded by severity from 0-4 following the Coronary Artery Disease Reporting and Data System (CAD-RADS) (Kumar and Bhatia (2022)). An alternative goal is to simply detect the presence/absence of vessel occlusion. The patients went through three non-invasive tests: exercise electrocardiogram, thallium scintigraphy, and cardiac fluoroscopy. The dataset comprises 66 features, from demographic information, *Age* and *Sex*, risk factors, family history, diabetes, cholesterol, and smoking history. Personal identifiable data, such as the name, ID and social security number were excluded.

During preprocessing, the dataset was substantially reduced to 581 samples and 38 features, due to a high rate of missing values. To grant better results, the task under analysis is the binary identification of coronary disease. Because FNs sustain a higher risk for the patients than FPs, the preferred performance parameter is recall (Recall or True Positive Rate (TPR)). The prevalence of the disease in the processed dataset is 62.9%.

## 3 Results and Discussion

The proposed counterfactual bias evaluation framework was validated on the two primary use cases, leveraging the Counterfactual Confusion Matrix (CCM) and the derived metrics. Additional use cases and their detailed analyses can be found in Appendix F.

A key aspect of our research lies in the framework's ability to deepen model evaluation, revealing biases potentially overlooked by conventional approaches. While standard metrics provide valuable insights into fairness concerns, they may fail to capture nuanced biases under the counterfactual setting. Our case studies aim to provide a comprehensive fairness evaluation in ML models, uncovering hidden biases and proving the effectiveness of our approach in highlighting unnoticed inequalities in complex real-world scenarios. We will focus on key metrics to present our empirical findings. Each use case is unique and best described with its most prominent metrics in terms of deviation from the ideal. As different models are compared in each experiment, the evolution of these values is also noted. To better guide the reader, the values mentioned in the text body, considered to be the most descriptive, are highlighted in bold in the correspondent tables.

### 3.1 CardioFollow.AI

In ML, bias is typically linked to fairness, particularly regarding protected attributes. However, analysing other features may also yield critical insights into the model's behaviour. This section discusses bias analysis for the smoking status feature. For reference, the reader may find the results for the sensitive feature *Sex* in Appendix F.2.4.

Smoking is a well-established risk factor that increases the probability of several lung and heart diseases (Gallucci et al. (2020)). As such, it seems reasonable to assume that the model would perform relatively well for individuals exhibiting this risk factor. However, the model underperforms for smokers, hinting at a dataset representational bias. As shown in Table 3, the TPR for the testing subset of non smokers is 74.6% and 71.4% for smokers.

In this use case, different sets of CFs were used. The base model mirrors the architecture described in Section 2.5.1, noting that the sensitive feature is not included in the model.

### 3.1.1 Detecting Bias

**CFs generated without Domain Knowledge**

Since smoking is correlated with most of the complications and preconditions, it seems reasonable to allow all the features to be adjusted when generating CFs. While $EOdds = 3.2p.p.$ ($TPR_{NS} = 74.6\%$ - $TPR_S = 71.4\%$) is not strictly significant, the resulting ECCM displayed in Figure 8 reveals a slight bias, associating non smokers with positive outcomes (post-surgery complications), supported by a $TNSR_{S \to NS} = 22.8\%$. Simultaneously, smokers are more linked to negative outcomes (uneventful recoveries), evidenced by $TPSR_{NS \to S} = 13.7\%$.

**CFs generated by removing correlation with the *Sex* feature**

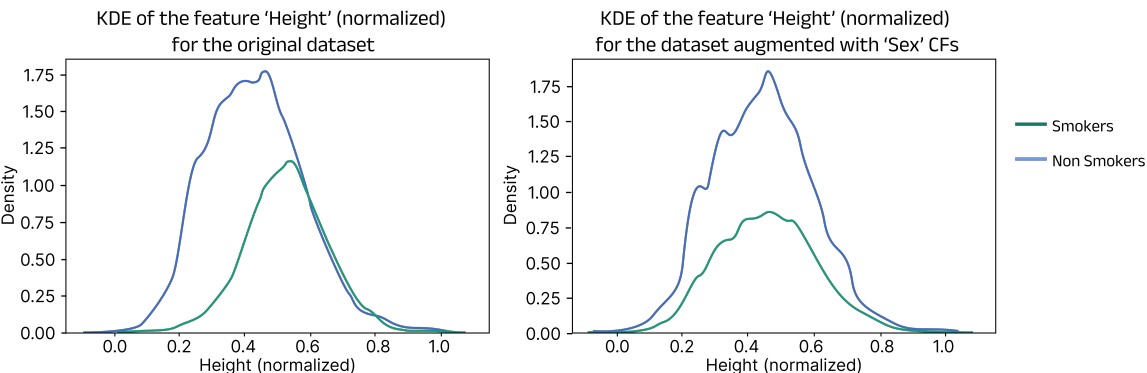

Figure 6: Kernel Density Estimation of the normalised *Height* of smokers and non smokers in the original dataset(Left) and in the dataset augmented with *Sex* CFs(Right).

A preliminary analysis found that *Smoking* is highly correlated with male patients in this dataset (88.0% of smokers are male). Moreover, the PDF of the patient's height is skewed to higher values in relation to non smokers (see Figure 6), which can influenced by the former correlation with sex. While height is not strictly related to smoking and could be easily excluded from the CF generation process, there are features related to both sex and smoking habits. For example, one side effect of smoking is the loss of appetite, which leads to smokers having on average a lower Body Mass Index (BMI). Weight can also be tied to the individual's sex, with men being generally heavier. For this reason, when switching from smoker to non smoker, because most smokers are men, whereas non smokers have approximately the same sex representation (52.3% female and 47.7% male), the tendency is to decrease the weight when the opposite would be closer to a real-world scenario. With these considerations, we balanced the representation by augmenting the dataset with *Sex*-based CFs, before generating the CFs regarding *Smoking*. These auxiliary CFs were generated only allowing changes in the features that should be related to the patient's sex, as described in Appendix F.2.4.

Table 2: Performance and Counterfactual Metrics referring to the base model trained on the CardioFollow.AI data, considering the sensitive feature *Smoking* with different sets of CFs.

| | CFs without decorrelation | | | Decorrelated CFs | | |
|---|---|---|---|---|---|---|
| MCC (%)↑ | 12.1 | 13.0 | 11.8 | 12.1 | 13.0 | 11.8 |
| TPR (%)↑ | 73.5 | **71.4** | **74.6** | 73.5 | 71.4 | 74.6 |
| TNR (%)↑ | 50.2 | 52.8 | 49.0 | 50.2 | 52.8 | 49.0 |
| SR (%)↓ | 10.4 | 14.6 | 8.4 | 14.6 | 28.8 | 7.6 |
| CMCC (%)↑ | 79.1 | 72.6 | 84.1 | 71.1 | 51.1 | 85.3 |
| PSR (%)↓ | 9.2 | 24.7 | 1.1 | 20.5 | 55.3 | 2.4 |
| NSR (%)↓ | 11.6 | 4.0 | 15.0 | 9.0 | 1.4 | 12.4 |
| TPSR (%)↓ | 9.8 | 2.6 | **13.7** | 8.0 | 2.6 | 10.9 |
| TNSR (%)↓ | 8.6 | **22.8** | 1.2 | 20.2 | **54.5** | 2.4 |
| FPSR (%)↓ | 11.8 | 4.2 | 15.2 | 9.1 | 1.2 | 12.6 |
| FNSR (%)↓ | 26.4 | 66.1 | 1.0 | 28.9 | 72.6 | 1.0 |
| RMSCD ↓ | 0.046 | 0.058 | 0.039 | 0.059 | 0.090 | 0.036 |
| JSCD ↓ | 0.038 | 0.038 | 0.027 | 0.048 | 0.047 | 0.026 |
| | **Total** | $S_{\rightarrow}NS$ | $NS_{\rightarrow}S$ | **Total** | $S_{\rightarrow}NS$ | $NS_{\rightarrow}S$ |

Figure 7: ECCM for the base model trained on the CardioFollow.AI data, considering the *Smoking* as sensitive feature with different sets of CFs. Left: CFs generated without removing correlation with the feature *Sex*; Right: CFs generated by removing correlation with the feature *Sex*.
$Y$ - Ground Truth (0 for *Uneventful Recoveries* and 1 for *Post-surgery Complications*); $G$ - Group ($S$ for *Smoker* and $NS$ for *Non Smoker*); $\hat{Y}$ - Prediction for the factual(original) sample; $\hat{Y}_{CF}$ - Prediction for the CF sample.

Statistically, combining the original training set with the *Sex*-base CFs, aids in eliminating correlations between sex and every feature that was not included in the generation process. As smoking is part of the non-included features, there is now an equivalent female smoker for every real male smoker. This process aligned the mean height values for smokers and non-smokers, as depicted in Figure 6. Subsequent analysis with this refined set of CFs showed an increased tendency to associate non-smokers to positive outcomes ($TNSR_{S\rightarrow NS} = 54.5\%$).

### 3.1.2 MITIGATING BIAS

We assessed the impact of CF augmentation on performance and bias, focusing on CFs decorrelated from *Sex*. A parallel experiment using CFs without this decorrelation is described in Appendix F.2.5.

**Data Augmentation using CFs without Selection**
A first experiment with all generated CFs revealed, as shown in Table 3, reduced switches from $TN_S$ to $FP_{NS}$ ($TNSR_{S \to NS} = 43.1\%$, $TPSR_{NS \to S} = 2.1\%$) but an aggravated TPR disparity between groups. This suggests a model seemingly more counterfactually fair, whereas it is still not able to reliably predict complications for smokers.

**Data Augmentation using CFs with Selection**
Since *Smoking* is a risk factor, intuitively, there are some logical implications that can be drawn. For instance, it is expected that if a non-smoker had complications, then if this patient smoked, they would also have complications due to the added risk only impairing their recovery. Similarly, if a smoker did not have complications post-surgery, then if they did not smoke, they would most likely not have complications as well. However, the remaining scenarios are more challenging to deduce the result. This hypothesis was tested by conducting augmentation with only selected CFs that obey these rules. This method essentially teaches the model about the added risk of smoking. However, an increased risk does not equal certainty, and thus not all CFs should be included, otherwise, the model would overfit on this feature. Some experiments led to the conclusion that, in this use case, a fraction of 10% allows to mitigate some of the existing CF bias, while increasing the performance for positive outcomes for smokers ($TPR_S = 71.9\%$) and non-smokers ($TPR_{NS} = 75.7\%$). The PredP remained low at $3.8p.p.$. The model's overall performance also improved slightly to positive outcomes ($TPR = 74.3\%$), but decreased for negative outcomes to a Specificity or True Negative Rate (TNR)= 49.5%. These metrics are summarised in Table 3.

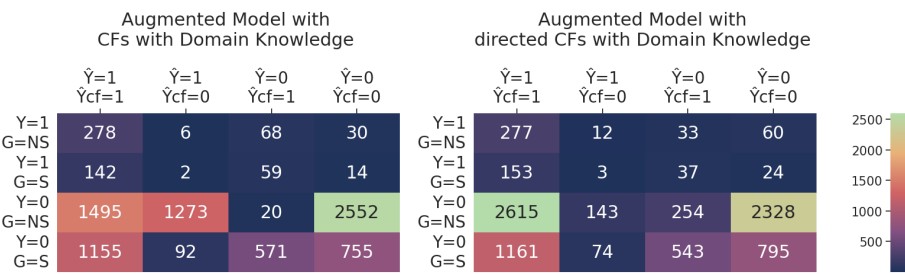

Figure 8: ECCM for the base model trained on the CardioFollow.AI data, considering the *Smoking* as sensitive feature, and using data augmentation with CFs generated by removing correlation with the feature *Sex*. Left: Model augmented with all the CFs; Right: Model augmented with directed CFs.
$Y$ - Ground Truth (0 for *Uneventful Recoveries* and 1 for *Post-surgery Complications*); $G$ - Group ($S$ for *Smoker* and $NS$ for *Non Smoker*); $\hat{Y}$ - Prediction for the factual(original) sample; $\hat{Y}_{CF}$ - Prediction for the CF sample.

Overall, the consistency of the predictions increased ($CMCC = 74.8\%$ vs $71.1\%$ - base model vs $51.7\%$ - augmentation with all the CFs). Following the risk factor intuition introduced before, we focus on the most relevant metrics suggesting reduced bias compared to the base model: $TPSR_{NS \to S}$ - decreased from $10.9\%$ to $4.2\%$, $TNSR_{S \to NS}$ - reduced from $54.5\%$ to $40.6\%$, and regarding the other direction, where $FNSR_{NS \to S}$ increased from $1.0\%$ to $35.5\%$, and $FPSR_{S \to NS}$ slightly rose from $1.2\%$ to $6.0\%$.

The $FNSR_{NS \to S} = 35.5\%$ indicates the model may associate non smokers with not having complications, which in turn is countered by a still considerable $TNSR_{S \to NS} = 40.6\%$. There is a trade-off between these metrics. Nevertheless, given the overall improvements, we consider the augmentation successful in reducing model bias. In relation to augmentation without CFs selection, these metrics improve with the exception of a slight increase in $TPSR_{NS \to S}$ from $2.1\%$ to $4.1\%$.

Table 3: Performance and Counterfactual Metrics for the base model trained on the CardioFollow.AI data, considering the sensitive feature *Smoking*, and using data augmentation with CFs generated by removing correlation with the feature *Sex*.

| | | | | Augmentation Method | | | | | |
| | Base Model | | | | All CFs | | | Directed CFs | |
|---|---|---|---|---|---|---|---|---|---|
| MCC (%)↑ | 12.1 | 13.0 | 11.8 | 10.6 | 9.6 | 11.3 | 12.2 | 12.8 | 12.0 |
| TPR (%)↑ | **73.5** | 71.4 | 74.6 | 71.5 | **66.4** | **74.4** | **74.3** | **71.9** | **75.7** |
| TNR (%)↑ | **50.2** | 52.8 | 49.0 | 49.3 | 51.6 | 48.2 | **49.5** | 52.0 | 48.4 |
| CMCC (%)↑ | 71.1 | 51.1 | 85.3 | 51.7 | 52.2 | 58.4 | 74.8 | 56.8 | 84.5 |
| TNSR (%)↓ | 20.2 | **54.5** | 2.4 | 15.2 | **43.1** | 0.8 | 20.3 | **40.6** | 9.8 |
| FNSR (%)↓ | 28.9 | 72.6 | **0.0** | 74.3 | 80.8 | **69.4** | 45.5 | 60.7 | **35.5** |
| TPSR (%)↓ | 8.0 | 2.6 | **10.9** | 1.9 | 1.4 | **2.1** | 3.4 | 1.9 | **4.2** |
| FPSR (%)↓ | 9.1 | 1.2 | 12.6 | 34.0 | 7.4 | 46.0 | 5.4 | 6.0 | 5.2 |
| RMSCD ↓ | 0.059 | 0.090 | 0.036 | 0.155 | 0.110 | 0.173 | 0.063 | 0.095 | 0.038 |
| JSCD ↓ | 0.048 | 0.047 | 0.026 | 0.139 | 0.075 | 0.139 | 0.048 | 0.063 | 0.031 |
| | **Total** | $S \to NS$ | $NS \to S$ | **Total** | $S \to NS$ | $NS \to S$ | **Total** | $S \to NS$ | $NS \to S$ |

This experience presents some of the challenges of handling datasets with underrepresented groups and highlights the need for domain knowledge in generating CFs. Furthermore, it illustrates that even if the sensitive feature *Smoking* is not included in the model training, the model can still yield different predictions for plausible CFs through correlated features.

### 3.1.3 FAIRNESS-PERFORMANCE TRADE-OFFS

The conducted experiment details the statistical alterations resulting from data augmentation with CFs, exploring the aspects relating to group and counterfactual fairness. This section will reflect on the trade-offs presented by 'optimising' fairness and the repercussions on performance.

Using group fairness metrics as an objective measure of fairness while keeping in mind the improvement in counterfactual metrics described in previous sections, the presented comparison focuses on classic performance metrics and their parity. Since the dataset is unbalanced, we focused on MCC, TPR and TNR, presenting ACC and Precision or Positive

Predicted Value (PPV) as valuable in terms of variation from the base model to the model achieved with the most successful mitigation strategy for each conducted experiment.

Table 4: Performance Metrics for: the base model trained on the CardioFollow.AI data and trained on the data augmented with directed CFs, considering the sensitive feature *Smoking* (Section 3.1).

| | CardioFollow.AI | | | | |
|---|---|---|---|---|---|
| ACC(%)↑ | **51.9** | **54.2** | **50.7** | 3.5 | **Base** |
| | 51.3 | 53.5 | 50.2 | **3.3** | **Aug.** |
| **Variation(p.p.)** | - 0.6 | - 0.7 | - 0.5 | - 0.2 | |
| MCC(%)↑ | 12.1 | 13.0 | 11.8 | 1.2 | **Base** |
| | 12.2 | 12.8 | 12.0 | 0.8 | **Aug.** |
| **Variation(p.p.)** | 0.1 | - 0.2 | 0.2 | - 0.4 | |
| TPR(%)↑ | 73.5 | 71.4 | 74.6 | **3.2** | **Base** |
| | **74.3** | **71.9** | **75.7** | 3.8 | **Aug.** |
| **Variation(p.p.)** | 0.8 | 0.5 | 1.1 | 0.6 | |
| TNR(%)↑ | **50.2** | **52.8** | **49.0** | 3.8 | **Base** |
| | 49.5 | 52.0 | 48.4 | **3.6** | **Aug.** |
| **Variation(p.p.)** | - 0.7 | - 0.8 | - 0.6 | - 0.2 | |
| PPV(%)↑ | **10.0** | **11.3** | **9.5** | 1.9 | **Base** |
| | **10.0** | 11.2 | **9.5** | **1.8** | **Aug.** |
| **Variation(p.p.)** | 0.0 | - 0.1 | 0.0 | - 0.1 | |
| | **Total** | $S \rightarrow NS$ | $NS \rightarrow S$ | **‖Parity‖** | |

When delving into fairness and bias mitigation, a critical discussion is whether, by attempting to mitigate bias, we are actively interfering and prejudicing the 'privileged' group in question, or, in some cases, every group at different rates. On the extreme, it may be asked if by attempting to solve fairness concerns, we are enabling the model to achieve worse results in favour of an idealised 'equality'. Regarding these concerns, we were conscious of the effect of our attempts to mitigate bias regarding performance fluctuations. Although trade-offs are expected and sometimes unavoidable, our efforts were guided by optimising both performance and fairness. In Table 4, we verify that augmentation with CFs did not particularly deteriorate performance metrics, keeping the values within close range. The best performance for each metric, before or post-mitigation, is highlighted in bold.

The results for the CardioFollow.AI usecase show a slight improvement in TPR, by $0.8 p.p.$, with a decrease in TNR by $0.7 p.p.$, alongside an improvement in counterfactual metrics. The accuracy also decreased by $0.7 p.p.$, but the performance was maintained for MCC, which accounts for class imbalance. In terms of group fairness, the disparity in TPR slightly increased, noting that the metric improved for both groups, $EOpp = 3.2 \nearrow 3.8 \ p.p.$. In contrast, the disparity among TNR slightly decreased $PredEq = 3.8 \searrow 3.6 \ p.p.$.

## 3.2 Heart Disease Dataset

### 3.2.1 Comparing CF generation methods

Following the architecture presented by Black et al. (2020) and the provided code (Yeom (2020)), this section compares their approximate method to optimal transport adapting

Wasserstein GANs, with our controlled generation process in terms of time efficiency and applicability. We note that we could not successfully train the GAN models for the CardioFollow.AI dataset, within the time budget available. The process did not result in appropriate subgroup changes, needed for our framework (Figure E.2).

**Time Efficiency**    To evaluate the time efficiency of the generation processes, the methods were run ten times for the Heart Disease dataset, described in Section 2.5.2. The public code requires that a GAN is trained for each directed combination of groups. As such, considering the sensitive feature *Sex*, it was necessary to train a GAN to generate 'males' from 'females' and another for the mirrored scenario. Given the objectives of our framework, the label of the CFs should remain the same. To ensure that, the data should be further sectioned by label, thus requiring training four distinct GANs. To mirror our process of generating plausible CFs, we trained another set of GANs with the plausible features.

Our method takes on average $16.69 \pm 0.13$ seconds to run whereas the combination of the four GANs takes on average $232.26 \pm 28.04$ seconds (approximately 14 times slower), granted that only CPU processing was used in these experiments[1].

If we generate CFs accounting for domain knowledge, only allowing changes in specific features, then our average run time is decreased to only $0.98 \pm 0.01$ seconds. Still, the combination of the four GANs trained on the plausible set of features took on average $182.72 \pm 5.19$ seconds. Our process can be further improved using multiprocessing since the generation of CFs is independent for each sample. The generative approach can run on GPU and improve training time. Moreover, depending on the dataset, the results may vary. In this particular instance, 500 epochs were needed to achieve proper representation. Additionally, we used the same architecture for the four GANs, but in practice, it could be challenging to incur in this route.

**Comparing Counterfactuals**    To study the CFs, we follow a similarity criterion between factuals and counterfactuals, calculating the cosine similarity for each pair. Additionally, we used dimension reduction methods and clustering to identify which CFs differ between our method and the GAN-based generation. The GAN-based approach keeps the pairs closer, with an overall median of 0.892, whereas our method's median stands at 0.846 (see Figure E.1 for a boxplot). Nevertheless, minimal changes may fall short from achieving proper CFs for our intended use. For instance, sensitive subgroups may have some overlap for a given feature subspace, which may signify different things for the group. For example, men and women can have the same body fat percentage but belong to different parts of the spectrum within their group since women typically have higher essential fat Bredella (2017). Here, a minimal change would fail to account for this scenario. Our method, based on statistical tools and domain knowledge considers these scenarios. The GAN-based method ideally avoids this by matching whole populations, but since it is not deterministic, it may still fall into this error. On the other hand, GANs' ability to learn inter-feature relations may contribute to more realistic CFs, a feature that may be a double-edged sword if there are undesirable or over-represented patterns in the training data.

---

1. **Machine Specs**
   **Processor** : 12th Gen Intel(R) Core(TM) i7-1255U 1.70 GHz
   **RAM** : 16,0 GB

To further study the differences between CFs results, we used Uniform Manifold Approximation and Projection (UMAP) to reduce dimensionality, applied to the set of plausible features, as the remaining features are unchanged. In the reduced space we started by mapping the trajectories from original samples to their counterfactual counterparts (see Figure E.1). Then, we used K-Neighbors Clustering to assign samples to eight different clusters. Finally, we calculated histograms for various features to find those with greater differences between the generation methods for different cluster assignments. As an example, we highlight the feature *restecg* that measures anomalies in the ST-T segment. This measure's efficacy has been reported as different for men and women, where the risk factor is reportedly more reliable for men than women (Elhendy et al. (1999)) [2]. We can thus infer that the value of this feature should vary when generating CFs. Our method made such changes in several instances, whereas the GAN-based method did not change this value for any CF (see Figure E.1). While the effective quality of the CFs may not be easily assessed, this example showcases a potential flaw when generating CFs with GANs as it may miss subtle patterns that can have a greater impact in the specific application. Our method, while with space for improvement, has the benefit of more controlled outputs.

### 3.2.2 Detecting Bias

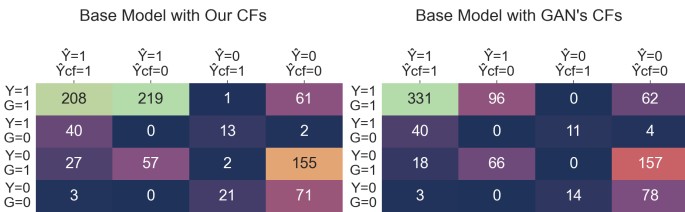

Figure 9: ECCM for the base model trained on the Heart Disease dataset, considering the sensitive feature *Sex* for CFs generated using a GAN or our method allowing changes in plausible features.
$Y$ - Ground Truth (0 for *Free from CAD* and 1 for *Signs of CAD*); $G$ - Group ($M$ for *Male* and $F$ for *Female*); $\hat{Y}$ - Prediction for the factual(original) sample; $\hat{Y}_{CF}$ - Prediction for the CF sample.

To evaluate the congruence and similarity between our CFs and the generated by the GAN, we trained the model and evaluated it in two phases as for the other experiments. Thus, using the defined plausible CFs for each generation process, we drew the ECCM for a base model and a model trained on the data augmented with CFs. Starting with the base model, we verify that the overall interpretation is similar (Figure 9).

The model seems to be prone to associate females with not having CAD marked by high $NSR_{M\to F}$ and $PSR_{F\to M}$ (see Table 5). However, comparatively to the CFs generated with the GAN($NSR_{M\to F} = 31.7\%$ and $PSR_{F\to M} = 23.4\%$), our CFs accuse more pronounced tendencies, ($NSR_{M\to F} = 54.0\%$ and $PSR_{F\to M} = 31.8\%$). Note that the performance metrics are the same. Only the counterfactual metrics vary since it is the same model. The

---

2. The Heart Disease dataset dates to 1988, supporting this reference that describes issues prior to 1999.

counterfactual metrics are on pair with group fairness metrics marked by higher TPR for men and higher TNR for women.

Table 5: Performance and Counterfactual metrics for base model trained on the Heart Disease dataset, considering the sensitive feature *Sex* for CFs generated using a GAN or our method allowing changes in the 'plausible' features.

| | | | | | Our Method | | | GAN | | |
|---|---|---|---|---|---|---|---|---|---|---|
| MCC (%)↑ | 60.3 | 74.1 | 53.8 | CMCC (%)↑ | 39.1 | 61.7 | 43.6 | 61.0 | 69.6 | 62.7 |
| ACC (%)↑ | 81.4 | 88 | 80 | SR (%)↓ | 35.6 | 22.7 | 38.2 | 21.3 | 16.7 | 22.2 |
| TPR (%)↑ | 85.8 | 72.7 | 87.3 | PSR (%)↓ | 11.3 | **31.8** | 1.4 | 7.7 | **23.4** | 0.0 |
| TNR (%)↑ | 74.1 | 96.8 | 65.1 | NSR (%)↓ | 49.8 | 0.0 | **54.0** | 29.2 | 0.0 | **31.7** |
| | **Total** | *F* | *M* | | **Total** | $F{\to}M$ | $M{\to}F$ | **Total** | $F{\to}M$ | $M{\to}F$ |

### 3.2.3 MITIGATING BIAS

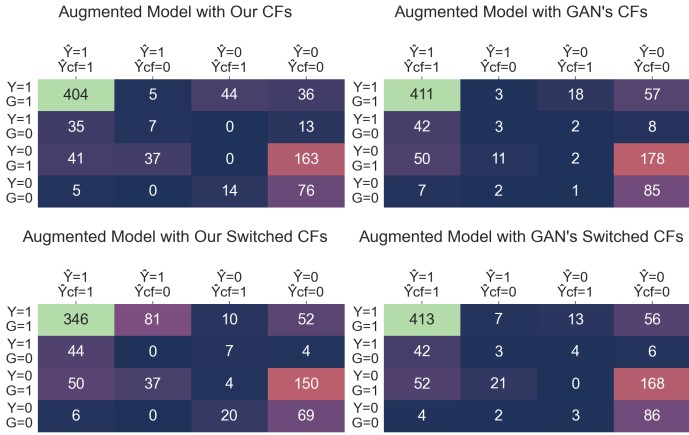

Figure 10: ECCM for the base model trained on the Heart Disease dataset, considering the sensitive feature *Sex* for CFs generated using a GAN-based method or our method allowing changes in plausible features.
$Y$ - Ground Truth (0 for *Free from CAD* and 1 for *Signs of CAD*); $G$ - Group ($M$ for *Male* and $F$ for *Female*); $\hat{Y}$ - Prediction for the factual(original) sample; $\hat{Y}_{CF}$ - Prediction for the CF sample.

To evaluate the effectiveness of the generated CFs in mitigating bias, we experimented with augmentation using all or only the switched CFs for each generation method. Results are presented in Table 6. Since the CFs for both approaches are different, the CF metrics are not comparable. Instead, we analyse the individual impact of augmentation versus the original values.

Both sets of CFs were effective in mitigating some bias while maintaining overall performance. For the CFs generated with the proposed method, we verify that augmentation

Table 6: Performance and Counterfactual Metrics referring to the base model trained on the Heart Disease dataset augmented with CFs generated using a GAN or our method allowing changes in plausible features, considering the sensitive feature *Sex*.

| | Our Method | | | | | | GAN-based (Black et al. (2020)) | | | | | |
| | All CFs | | | Switched CFs | | | All CFs | | | Switched CFs | | |
|---|---|---|---|---|---|---|---|---|---|---|---|---|
| ACC (%)↑ | 80.0 | 88.0 | 78.4 | 81.1 | 88.7 | 79.6 | 82.4 | 87.3 | 81.4 | 82.0 | 89.3 | 80.5 |
| MCC (%)↑ | **57.9** | 73.9 | 51.2 | **59.7** | 75.3 | 52.8 | 63.0 | 72.6 | 58.5 | 62.0 | 76.8 | 55.8 |
| TPR (%)↑ | 82.9 | 76.4 | 83.6 | **86.6** | 80.0 | 87.3 | 84.4 | 81.8 | 84.7 | **85.5** | 81.8 | 85.9 |
| TNR (%)↑ | 75.3 | 94.7 | 67.6 | **72.3** | 93.7 | 63.9 | 79.2 | 90.5 | 74.7 | **76.5** | 93.7 | 69.7 |
| CMCC (%)↑ | **74.4** | 69.1 | 73.4 | **63.8** | 68.8 | 64.9 | 90.0 | 88.4 | 89.7 | 87.4 | 82.4 | 87.5 |
| SR (%)↓ | 12.2 | 14.0 | 11.8 | 18.1 | 18.0 | 18.1 | 4.8 | 5.3 | 4.7 | 6.0 | 8.0 | 5.6 |
| PSR (%)↓ | 16.8 | 13.6 | 18.1 | 13.0 | **27.0** | 6.5 | 6.6 | 3.1 | 7.8 | 6.0 | **7.1** | 5.5 |
| NSR (%)↓ | 9.2 | 14.9 | 8.6 | 20.9 | 0.0 | **23.0** | 3.6 | 9.3 | 2.9 | 6.1 | 9.8 | **5.7** |
| | **Total** | $F{\to}M$ | $M{\to}F$ | **Total** | $F{\to}M$ | $M{\to}F$ | **Total** | $F{\to}M$ | $M{\to}F$ | **Total** | $F{\to}M$ | $M{\to}F$ |

with all CFs was more effective in minimising switches, ($CMCC = 74.4\%$ vs 63.8% - Augmentation with switched CFs vs 39.1% - Base Model). However, adding only switched CFs achieved greater overall performance ($MCC = 59.7\%$ vs 57.9% - Augmentation with all CFs vs 60.3% - Base Model). The augmentation with switched CFs was able to mitigate the $PSR_{F{\to}M}$ from 31.8% to 27.0% and $NSR_{M{\to}F}$ from 54.0% to 23.0%. As for CFs generated with the GANs, the augmentation was also effective. Adding only switched CFs, the $PSR_{F{\to}M}$ decreased from 23.4% to 7.1% and $NSR_{M{\to}F}$ from 31.7% to 5.7%. Concerning group fairness, adding the GANs' CFs was more effective in mitigating group bias, with $EOpp = 4.1p.p.$ vs $7.3p.p.$ - Augmentation with our switched CFs vs $14.6p.p.$ - Base model; $PredEq = 24.0p.p.$ vs $30.2p.p.$ - Augmentation with our switched CFs vs $31.7p.p.$ - Base model. In terms of performance, adding our CFs achieved slightly better results for positive outcomes ($TPR = 86.6\%$ vs 85.5% - Augmentation with GANs' switched CFs vs 85.8% - Base model) while GAN-based CF augmentation was more efficient for predicting negative outcomes ($TNR = 76.5\%$ vs 72.3% - Augmentation with our switched CFs vs 74.1% - Base model).

## 4 Conclusions and Future Work

The proposed framework for bias evaluation extends the concept of Counterfactual Fairness, aggregating individual metrics into a more comprehensive group-based analysis. Counterfactual sample generation is a key step, and this work introduces a methodology to create CFs based on training set statistical distributions to increase plausibility. As a result, not only the sensitive feature but other, possibly correlated features, are also adjusted in the counterfactual scenario. Notably, our framework can be applied even when omitting the sensitive feature from the learning stage, allowing to still evaluate model bias. Additionally, domain knowledge can be leveraged to reach a more realistic counterfactual setting (e.g., by only allowing a subset of features to be adjusted, even if others are - spuriously - statistically related). Our method is less strict than a purely causal model, required by other counterfactual fairness methods, while still providing insights on how to improve the model.

It is a user-friendly agnostic approach suitable as a first line for bias assessment. We note, however, the impact that the generation process has on interpreting model bias, as it may, by itself, introduce additional bias. Given this, when ensuing into fairness assessment it is important to clearly evaluate the dataset, identify relations and corroborate them through scientific evidence and align expectations with the fairness requirements. Even then, certain patterns, biased or not, are often missed, as shown for an alternative GAN-based approach for CFs generation. Our methodology calls for a conservative approach when selecting related features, acknowledging the context application and relevancy of said feature not only to the sensitive attribute but to the problem itself. While this may pose a challenge, we believe that data engineering and careful research are imperative when dealing with sensitive topics. Moreover, the proposed generation process can be more or less constricted, limiting the maximum depth, as well as adapting threshold flips. Through experimentation and validation of CFs, for instance, one can assess if they would be considered outliers in the new group, or observe changes in group fairness when using them for data augmentation. The method is suited for larger datasets as there are more points of reference for flips and the precision point for metrics also increases. More complex models with larger feature space may pose a challenge in preliminary steps when choosing the features and are more computationally demanding. Nevertheless, there is the benefit of less general features, though further testing would be required to assess the process efficiency.

By relying more superficially on existing representations, our approach ensures plausible changes, avoiding implausible inter- and extrapolations, thus providing a more reliable evaluation of model bias. Although not the core contribution of this work, we propose an alternative CF generation approach, compared to that of Black et al. (2020). Our approach offers finer control over the generation process to capture subtle patterns, and it excels in computational efficiency, avoiding training additional models. Still, if successfully trained, the generative approach may be used as an augmentation source to improve both performance and fairness.

Aiming to facilitate model bias analysis with an aggregated perspective, this work introduces a new take on the Confusion Matrix, tailored to a counterfactual setting. The Counterfactual Confusion Matrix (CCM) and its extended version ECCM provide clear and efficient means to assess the susceptibility of a model to changes in a specified sensitive attribute. As an agnostic instrument, it is extremely flexible to different tasks, and while it has only been demonstrated in binary classification tasks for binary sensitive features, it can be employed without loss of generalisation to categorical features and multiclass problems. The derived metrics offer valuable insights into the presence of bias and how it impacts specific subgroups. Moreover, the ECCM allows a more granular view on potential bias sources, such as the model's higher susceptibility when making Type-I errors. These insights could facilitate targeted bias mitigation.

We demonstrated the applicability and complementarity of our framework in real-world datasets. The findings support the need for a new perspective, as existing bias mitigation techniques focusing on specific metrics may inadvertently compromise other important fairness criteria. Furthermore, our work showcases the flexibility of using CFs for mitigation techniques, through different selection methods to ensure more credible results, supported by domain knowledge; and efficiency, by filtering the not consistent CFs.

In future research, we aim to enhance the generation process by integrating the benefits of generative models for improved robustness and incorporating Human-in-the-loop methodologies for added control. We also intend to explore alternative data sources, like real-world statistics to avoid local biases. The bias evaluation metrics will undergo a formal review, focusing on trade-off analysis akin to ROC curves. Moreover, we will investigate how to incorporate our findings into model fitting as constraints. Concerning bias mitigation, we will explore how to streamline the use of selectively curated CFs during learning to minimise data redundancy. Additionally, the framework's scope could extend to assess model robustness from different perspectives, such as improving out-of-distribution coverage, thereby enhancing decision-support systems to benefit society at large.

## Broader Impact Statement

This work introduces a novel framework for evaluating and mitigating bias in ML applications, extending the counterfactual setting to a group-based analysis. Our approach offers a new perspective in bias analysis, contributing to the field of AI fairness, while emphasising the critical role of domain knowledge in creating plausible CFs for evaluation and augmentation.

The potential misuse of the counterfactual generation process, if not grounded in domain expertise, can inadvertently introduce biases, oversimplifying complex societal issues. Thus, we highlight the importance of incorporating diverse, real-world knowledge and perspectives, to be improved in future research. As we stress that bias in ML is highly context-dependent, our framework should be integrated with other fairness metrics for a holistic analysis, rather than used in isolation. This will ensure the proposed methodology not only identifies bias but also accurately supports its interpretation, contributing to the development of equitable AI systems that are more inclusive, representative, and beneficial for society.

## Acknowledgments and Disclosure of Funding

This work was supported by European funds through the Recovery and Resilience Plan, via project "Center for Responsible AI", with identification number C645008882-00000055. We further thank the authors who made the datasets available for study, namely the Adult dataset (Kohavi and Becker (1996)), COMPAS(ProPublica (2017)), and Heart Disease (Asuncion and Newman (2007)), as well as the team from project CardioFollow.AI.

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

# Appendix A. Pseudo-Code of the Counterfactual Generation Process

---

**Algorithm 1** Counterfactual Generation

---

1: **procedure** GENERATECOUNTERFACTUALS(TrainingSet, Samples, PotentialFeatures=[BinaryFeatures, CategoricalFeatures, ContinuousFeatures], SensitiveFeature, minThreshold, maxDepth)
2:      Initiate $Samples_{CF}$ as a copy of $Samples$
3:      **for** each feature $f_i$ not in $BinaryFeatures$ **do**
4:          Calculate $P_g^y$ of $f_i$ in the $TrainingSet$ for each group $g$ in $SensitiveFeature$
5:          and label $y$
6:          **for** each sample $s_k$ in $Samples_{CF}$ **do**
7:              **if** $f_i$ is in $ContinuosFeatures$ **then**
8:                  Interpolate to find $v'_{k,i}$ in CDF $P_{g_{CF}}^y$ based on $v_{k,i}$ in $P_g^y$
9:              **else if** $f_i$ is in $CategoricalFeatures$ **then**
10:                 Find closest value $v'_{k,i}$ in CDF $P_{g_{CF}}^y$ based on $v_{k,i}$ in $P_g^y$
11:              **end if**
12:          **end for**
13:      **end for**
14:      **for** each sample $s_k$ in $Samples_{CF}$ **do**
15:          Flip value of $SensitiveFeature$
16:          $s_k = FlipBinaryFeatures(s_k, BinaryFeatures, TrainingSet_{label=y},$
17:          $SensitiveFeature, minThreshold, maxDepth)$
18:      **end for**
19:      **return** $Samples_{CF}$
20: **end procedure**
21: **procedure** FLIPBINARYFEATURES(Sample, BinaryFeatures, TrainingSet, SensitiveFeature, minThreshold, maxDepth, iter=0)
22:      Initiate $Sample_{CF}$ as a copy of $Sample$
23:      **for** each binary feature $f_i$ in $BinaryFeatures$ **do**
24:          **if** $iter < maxDepth$ **then**
25:          Calculate the difference in conditional probability of the value $v_i$ of $f_i$
26:          given the original, $g$, and new group, $g_{CF}$, of the $SensitiveFeature$
27:          $ProbDiff = |P(v_i|g) - P(vi|g_{CF})$
28:          **if** $ProbDiff > minThreshold$ **then**
29:              Flip the binary feature value
30:              Start new iteration
31:              $iter += 1$
32:              $Sample_{CF} = FlipBinaryFeatures(Sample_{CF}, BinaryFeatures - f_i,$
33:                  $TrainingSet[SensitiveFeature = g_{CF}], SensitiveFeature = f,$
34:                  $minThreshold, maxDepth, iter)$
35:              **end if**
36:          **end if**
37:      **end for**
38:      **return** $Sample_{CF}$
39: **end procedure**

---

## Appendix B. Extended Counterfactual Confusion Matrix (ECCM)- Additional Content

The broken down version of the ECCM retains the introduced properties, being easily converted into the ECCM, CCM and traditional CM. Its main suit is including all the required information for any (classification) performance or bias evaluation metric.

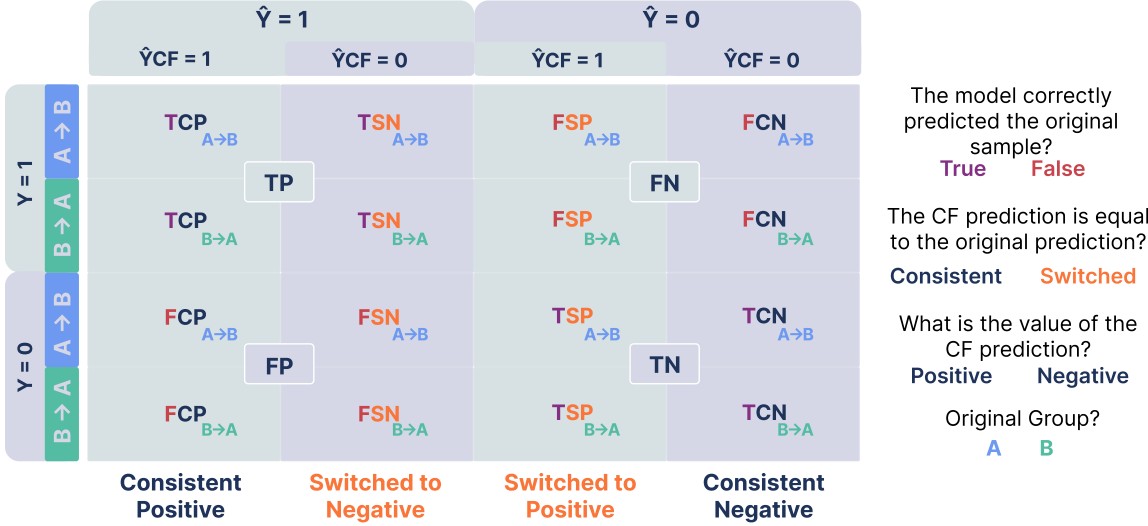

Figure 11: The Extended Counterfactual Confusion Matrix broken down into groups. $Y$ - Ground Truth; $\hat{Y}$ - Prediction for the factual(original) sample; $\hat{Y}_{CF}$ - Prediction for the CF sample.

### B.1 Edge Cases

The ideal ECCM represents a case where every CF is consistent with its original sample. Visually, this corresponds to having both the inner columns empty. However, an ideal ECCM does not necessarily correspond to an ideal CM, or vice-versa. An ideal CM corresponds to an ECCM with the FCP, False Switched Positives (FSP), False Consistent Negatives (FCN), and FSN cells empty. An ideal model in both criteria is represented by an ECCM which only has values in the TCP and True Consistent Negatives (TCN) cells. These scenarios are represented in Figure 12.

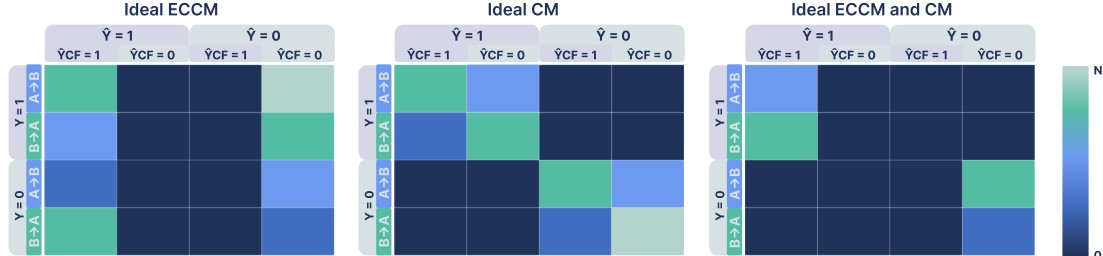

Figure 12: Edge cases of Extended Counterfactual Confusion Matrix: (1) for a not ideal model for performance but that is CF fair; (2) of an ideal model for performance (ideal CM), but not in terms of CF fairness; (3) for an ideal model both in terms of performance and CF fairness. Note that the colour gradation between blue and green is merely representative of possible data partitions between groups and labels, the main takeaway is these cells' non-null values.

## Appendix C. Method Scalability

The presented notation examples describe a generation and evaluation process initially drawn for binary classes and sensitive features. However, the process is easily adaptable to categorical sensitive features and multiclass problems. The main drawback is the increase of computational cost with the increase of CFs generated and possible increase in the complexity of the results analysis. However, it is a minor setback for a more thorough evaluation of fairness, only requiring extra care and ensuing additional steps of evaluation. Moreover, if the dataset sufficiently represents its subgroups to allow for these scenarios, the evaluation through CFs should still be effective.

### C.1 Counterfactual Generation

The generation process is suitable for categorical sensitive features through parallel processes, where each paired combination of subgroups is treated separately. In these situations, the number of generated CFs increases, leading to a higher computational cost. For a group of $S$ samples in a dataset with $G$ subgroups, the number of generated CFs is $(G - 1) \times S$.

The process is also suitable for multiclass problems given that the changes occur among the subgroup samples with the same label. Essentially, there is a parallel process to generate CFs for each label. If the dataset is big enough to support reliable multiclass classification, the generation should not suffer from the process. On the other hand, the severe under-representation of subgroups may require extra care when employing this process, as relying on data distributions may not be the most suitable strategy.

### C.2 Counterfactual Evaluation

The proposed metrics cover the analysis for binary sensitive features, although they can also be used for categorical ones. In this instance, one can approach the problem not only

pair by pair but also in a 'one-versus-all' strategy. For example, given three groups, $A$, $B$, and $C$, to study bias, it may be relevant to study the switches from $A \to B$ and $A \to C$, as well $A \to B \vee C$. Metrics that suggest bias against the original group, e.g. FPSR, are suited for the $A \to B \vee C$ case. In contrast, metrics that indicate bias against the new group, e.g. TPSR, are better suited for pair distinction, $A \to B$ and $C \to B$, or the inverse of the 'one-versus-all' strategy, $A \vee C \to B$.

Similar to the CM performance parameters, these metrics also apply to multiclass problems. Akin to sensitive features with multiple groups, it may be interesting to group outcomes for specific analyses. Considering three possible outputs $I$, $K$, and $J$, metrics that evaluate clear tendencies to one particular output, e.g. PSR, are suited for the 'one-versus-all' strategy, in this case, exploring the portion of samples that were predicted as $K$ or $J$ but switched to $I$. If the behaviour is attested from both $K$ and $J$ groups, it better supports the hypothesis that the model associates the new group with $I$ than if it only occurs for samples initially predicted as $K$, although both are important considerations.

## C.3 Counterfactual metrics cheat sheet

Table 7: Summary of the CF metrics and its interpretation extrapolated from the Counterfactual Confusion Matrix and its extended version.

Metrics derived from CCM

| | | | Group or Global Metric | | | | Parity (A - B) | | |
|---|---|---|---|---|---|---|---|---|---|
| CCM Metric | Range | Highest Bias Value | Intuition | Impaired Group | CM Analog | Range | Impaired Group | Group Fairness Analog |
| CR = 1 - SR | [0,1] | 0 | The model relies heavily on the distributions represented in the dataset for each group of the sensitive feature, indicating potential bias | New Group | ACC | [-1,1] | [-1,0[ A ]0,+1] B | Overall Accuracy Parity |
| PCR = 1 - NSR | [0,1] | 0 | The model associates the new group with negative outcomes, indicating bias | New Group | TNR | [-1,1] | [-1,0[ A ]0,+1] B | PredEq |
| NCR = 1 - PSR | [0,1] | 0 | The model associates the new group with positive outcomes, indicating bias | Original Group | TPR | [-1,1] | [-1,0[ A ]0,+1] B | EOpp |
| PCP = 1 - PSDR | [0,1] | 0 | The model associates the new group with positive outcomes, indicating bias | Original Group | PPV | [-1,1] | [-1,0[ A ]0,+1] B | PredP |
| CMCC | [-1,1] | -1 | The model relies heavily on the distributions represented in the dataset for each group of the sensitive feature | New Group | MCC | [-2,2] | [-2,0[ A ]0,+2] B | - |

Metrics derived from ECCM: Including the Ground Truth

| | | | Group or Global Metric | | | Parity (A - B) | |
|---|---|---|---|---|---|---|---|
| ECCM Metric | Range | Highest Bias Value | Intuition | Impaired Group | Range | Impaired Group |
| TPSR | [0,1] | 1 | The model is not able to predict correctly positive outcomes based on the group. It indicates the need for more variability in samples with positive ground truth for the new group | New Group | [-1,1] | [-1,0[ A ]0,+1] B |
| TNSR | [0,1] | 1 | The model is not able to predict correctly negative outcomes based on the group. It indicates the need for more variability in samples with negative ground truth for the new group | New Group | [-1,1] | [-1,0[ A ]0,+1] B |
| FPSR | [0,1] | 1 | The model is not able to predict correctly negative outcomes based on the group. It indicates the need for more variability in samples with negative ground truth for the original group | Original Group | [-1,1] | [-1,0[ B ]0,+1] A |
| FNSR | [0,1] | 1 | The model is not able to predict correctly positive outcomes based on the group. It indicates the need for more variability in samples with positive ground truth for the new group | Original Group | [-1,1] | [-1,0[ B ]0,+1] A |

## Appendix D. List Of Features CardioFollow.AI Dataset

Currently, there is a deployed model optimised with a minimum set of eight features comprising total time of stay (1), of that time: in intensive care (2); time to surgery (3) time from surgery until discharge (4); number of complications (5); time in bypass (6); implant size (7); and type of diabetes treatment (if any) (8). The time measurements are adequate proxies for inferring urgency and how the patient recovered, whereas the other features serve to assess the gravity of the occurrence. While other parameters are relevant for this task, due to the relatively small dataset for the total possible combinations of risk factors, procedures and complications, through careful experimentation with SHapley Additive exPlanations (SHAP) values, these are the features that paint the best overall scenario.

| Category | Variables |
|---|---|
| Demographic Information | Age, Gender, Height, Weight |
| Pre-operative Risk Factors | BMI, Body Surface Area (BSA), Smoking History, Diabetes, Hypertension, Hypercholesteremia |
| Existing Pathologies/ Dysfunctions | Renal Pathologies, Lung Pathologies, Vascular Pathologies, Neurological Pathologies/ Dysfunctions |
| Procedure Specifics | Heart Rhythm, Number of Diseased Coronary Vessels, Results from Cardiac Catheterisation |
| Procedure Context | State of the Patient, Urgency, Cause of Intervention, Type of Procedure, Procedure Specifications, Implant Type |
| Morbidity Scale Assessment | Complications, Morbidity Scale |
| Chronological Information | Duration of Procedure, Pre-operation Period, Post-operation Period, Time Since Last Consult |
| Target Variable | Occurrence or Absence of Post-surgery Health Complications |

Table 8: Lists of features of CardioFollow.AI Dataset

# Appendix E. Comparison between counterfactuals' generation approaches

## E.1 Heart Disease

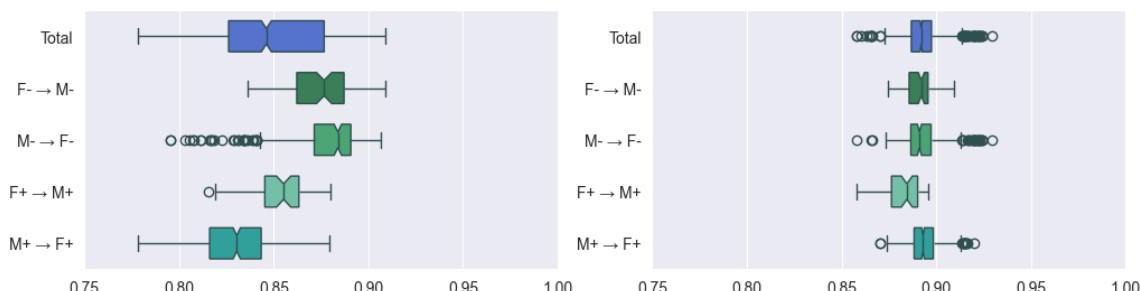

Figure 13: Cosine similarity between pairs of original samples from the Heart Disease dataset and their respective CFs generated with our method(left) or with a GAN(right) using filtered features, considering the sensitive feature *Sex*.

Table 9: Time performance for CF generation using a GAN or our method allowing changes in 'all' or only the 'plausible' features, trained on the Heart Disease dataset, considering the sensitive feature *Sex*.

| | GAN | | | | | | | | Our Method | |
| | All | | | | Plausible | | | | All | Plausible |
|---|---|---|---|---|---|---|---|---|---|---|
| 0 | 59.61 | 55.59 | 60.34 | 63.72 | 44.41 | 46.48 | 44.42 | 45.84 | 16.83 | 0.96 |
| 1 | 61.71 | 61.77 | 63.97 | 59.23 | 48.49 | 45.45 | 46.84 | 44.86 | 16.76 | 1.01 |
| 2 | 56.71 | 57.54 | 58.62 | 58.02 | 46.20 | 47.85 | 46.21 | 45.14 | 16.49 | 0.96 |
| 3 | 56.48 | 58.45 | 58.87 | 58.23 | 45.01 | 47.07 | 44.58 | 45.13 | 16.85 | 0.98 |
| 4 | 57.08 | 57.36 | 59.27 | 61.62 | 47.07 | 48.08 | 47.03 | 44.33 | 16.66 | 0.98 |
| 5 | 57.34 | 61.70 | 58.78 | 57.44 | 44.95 | 45.24 | 46.38 | 45.58 | 16.55 | 0.98 |
| 6 | 57.73 | 60.97 | 58.40 | 57.47 | 45.73 | 44.98 | 44.69 | 46.08 | 16.58 | 0.98 |
| 7 | 55.69 | 54.83 | 54.92 | 56.67 | 45.13 | 44.62 | 46.26 | 44.43 | 16.71 | 0.98 |
| 8 | 56.01 | 54.84 | 55.00 | 55.84 | 45.18 | 45.16 | 44.69 | 45.84 | 16.85 | 0.97 |
| 9 | 55.43 | 54.17 | 54.53 | 60.68 | 44.67 | 45.11 | 47.17 | 44.79 | 16.63 | 0.98 |
| | -F→-M | -M→-F | +F→+M | +M→+F | -F→-M | -M→-F | +F→+M | +M→+F | | |
| $\mu$ | 57.38 | 57.72 | 58.27 | 58.89 | 45.68 | 46.00 | 45.83 | 45.20 | 16.69 | 0.98 |
| | | **Total** | | 232.26 | | | **Total** | 182.72 | | |
| $\sigma$ | 1.94 | 2.93 | 2.88 | 2.44 | 1.26 | 1.27 | 1.11 | 0.61 | 0.13 | 0.01 |
| | | **Total** | | 28.04 | | | **Total** | 5.19 | | |

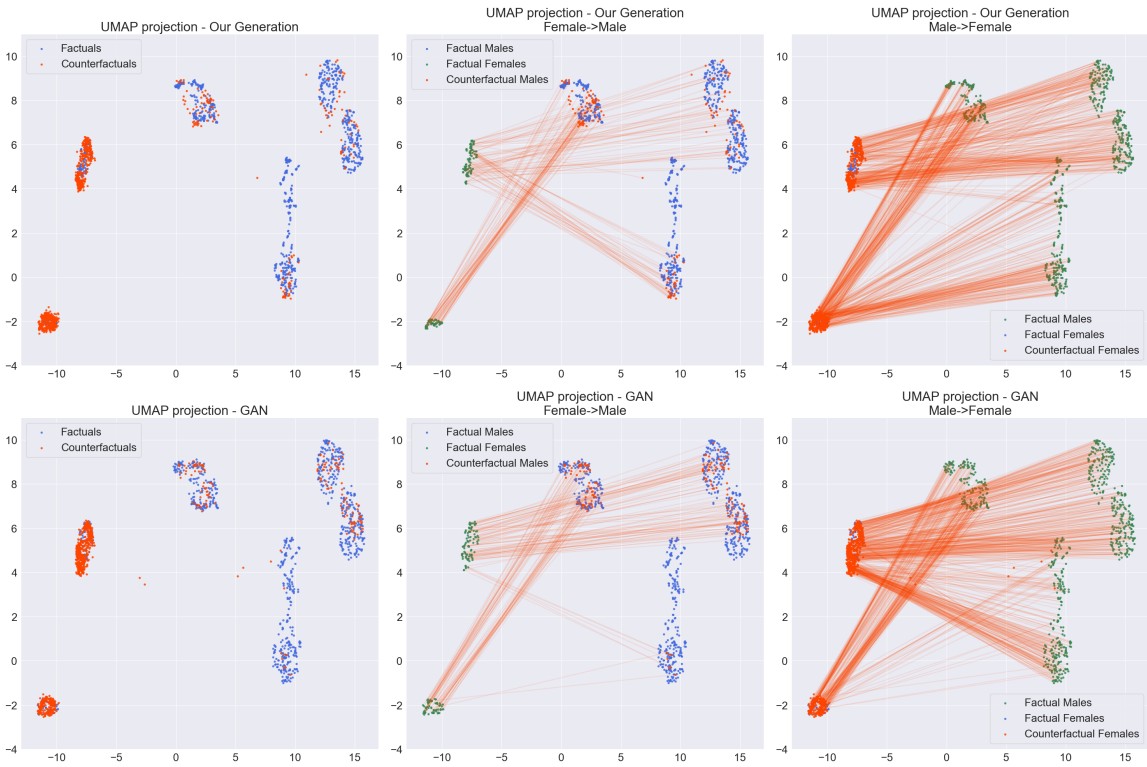

Figure 14: Samples distribution from the Heart Disease dataset in a two-dimensional space after applying UMAP. Mappings for CFs generated with our method (Top Row) and through GANs (Bottom Row), showcasing all CFs(Left) or sectioned by flip: Female→ Male (Middle); Male→ Female (Right).

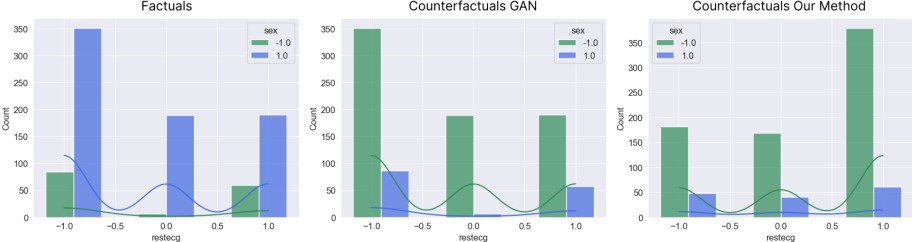

Figure 15: Histogram of the feature *restecg* for factuals(Left) and counterfactuals generated with GANs(Middle) and with our method(Right) for samples from the Heart Disease dataset, considering the sensitive feature *Sex*.
Green - Women; Blue - Men

Table 10: Performance and Counterfactual metrics for the base model trained on the Heart Disease dataset, considering the sensitive feature *Sex* for CFs generated using a GAN or our method allowing changes on only the 'plausible' features.

|  | Total | F | M |  | Our Method | | | GAN | | |
|---|---|---|---|---|---|---|---|---|---|---|
| MCC (%)↑ | 60.3 | 74.1 | 53.8 | CMCC (%)↑ | 39.1 | 61.7 | 43.6 | 61.0 | 69.6 | 62.7 |
| ACC (%)↑ | 81.4 | 88 | 80 | SR (%)↓ | 35.6 | 22.7 | 38.2 | 21.3 | 16.7 | 22.2 |
| TPR (%)↑ | 85.8 | 72.7 | 87.3 | PSR (%)↓ | 11.3 | 31.8 | 1.4 | 7.7 | 23.4 | 0.0 |
| TNR (%)↑ | 74.1 | 96.8 | 65.1 | NSR (%)↓ | 49.8 | 0.0 | 54.0 | 29.2 | 0.0 | 31.7 |
|  | **Total** | *F* | *M* | TPSR (%)↓ | 46.9 | 0.0 | 51.3 | 20.6 | 0.0 | 22.5 |
|  |  |  |  | FNSR (%)↓ | 18.2 | 86.7 | 1.6 | 14.3 | 73.3 | 0.0 |
|  |  |  |  | TNSR (%)↓ | 9.2 | 22.8 | 1.3 | 5.6 | 15.2 | 0.0 |
|  |  |  |  | FPSR (%)↓ | 65.5 | 0.0 | 67.9 | 75.9 | 0.0 | 78.6 |
|  |  |  |  | RMSCD ↓ | 0.385 | 0.275 | 0.404 | 0.239 | 0.228 | 0.241 |
|  |  |  |  | JSCD ↓ | 0.270 | 0.161 | 0.243 | 0.163 | 0.151 | 0.130 |
|  |  |  |  |  | **Total** | $F{\to}M$ | $M{\to}F$ | **Total** | $F{\to}M$ | $M{\to}F$ |

Table 11: Performance and Counterfactual Metrics referring to the base model trained on the Heart Disease dataset augmented with CFs generated using a GAN or our method allowing changes in plausible features, considering the sensitive feature *Sex*.

|  | Our Method | | | | | | GAN-based (Black et al. (2020)) | | | | | |
|---|---|---|---|---|---|---|---|---|---|---|---|---|
|  | All CFs | | | Switched CFs | | | All CFs | | | Switched CFs | | |
| ACC (%)↑ | 80.0 | 88.0 | 78.4 | 81.1 | 88.7 | 79.6 | 82.4 | 87.3 | 81.4 | 82.0 | 89.3 | 80.5 |
| MCC (%)↑ | 57.9 | 73.9 | 51.2 | 59.7 | 75.3 | 52.8 | 63.0 | 72.6 | 58.5 | 62.0 | 76.8 | 55.8 |
| TPR (%)↑ | 82.9 | 76.4 | 83.6 | 86.6 | 80.0 | 87.3 | 84.4 | 81.8 | 84.7 | 85.5 | 81.8 | 85.9 |
| TNR (%)↑ | 75.3 | 94.7 | 67.6 | 72.3 | 93.7 | 63.9 | 79.2 | 90.5 | 74.7 | 76.5 | 93.7 | 69.7 |
| CMCC (%)↑ | 74.4 | 69.1 | 73.4 | 63.8 | 68.8 | 64.9 | 90.0 | 88.4 | 89.7 | 87.4 | 82.4 | 87.5 |
| SR (%)↓ | 12.2 | 14.0 | 11.8 | 18.1 | 18.0 | 18.1 | 4.8 | 5.3 | 4.7 | 6.0 | 8.0 | 5.6 |
| PSR (%)↓ | 16.8 | 13.6 | 18.1 | 13.0 | 27.0 | 6.5 | 6.6 | 3.1 | 7.8 | 6.0 | 7.1 | 5.5 |
| NSR (%)↓ | 9.2 | 14.9 | 8.6 | 20.9 | 0.0 | 23.0 | 3.6 | 9.3 | 2.9 | 6.1 | 9.8 | 5.7 |
| TPSR (%)↓ | 2.7 | 16.7 | 1.2 | 17.2 | 0.0 | 19.0 | 1.3 | 6.7 | 0.7 | 2.2 | 6.7 | 1.7 |
| FNSR (%)↓ | 47.3 | 0.0 | 55.0 | 23.3 | 63.6 | 16.1 | 23.5 | 20.0 | 24.0 | 21.5 | 40.0 | 18.8 |
| TNSR (%)↓ | 5.5 | 15.6 | 0.0 | 9.9 | 22.5 | 2.6 | 1.1 | 1.2 | 1.1 | 1.2 | 3.4 | 0.0 |
| FPSR (%)↓ | 44.6 | 0.0 | 47.4 | 39.8 | 0.0 | 42.5 | 18.6 | 22.2 | 18.0 | 29.1 | 33.3 | 28.8 |
| RMSCD ↓ | 0.177 | 0.188 | 0.175 | 0.180 | 0.154 | 0.185 | 0.089 | 0.105 | 0.086 | 0.086 | 0.102 | 0.082 |
| JSCD ↓ | 0.124 | 0.184 | 0.113 | 0.117 | 0.088 | 0.106 | 0.066 | 0.098 | 0.060 | 0.062 | 0.091 | 0.055 |
|  | **Total** | $F{\to}M$ | $M{\to}F$ | **Total** | $F{\to}M$ | $M{\to}F$ | **Total** | $F{\to}M$ | $M{\to}F$ | **Total** | $F{\to}M$ | $M{\to}F$ |

## E.2 CardioFollow.AI

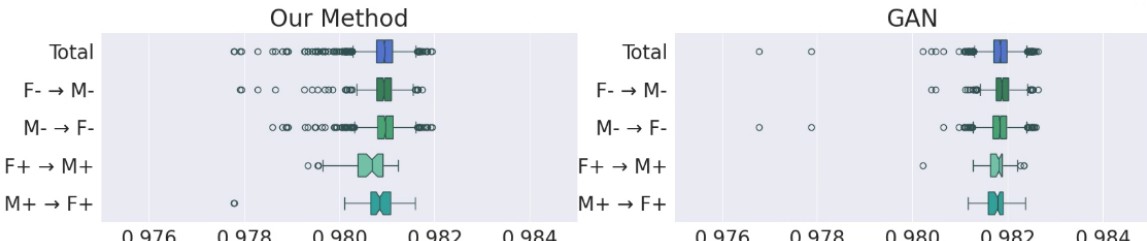

Figure 16: Cosine similarity between pairs of original samples from the Cardio.Follow.AI dataset and their respective CFs generated with our method(left) or with a GAN(right) using filtered features, considering the sensitive feature *Sex.*.

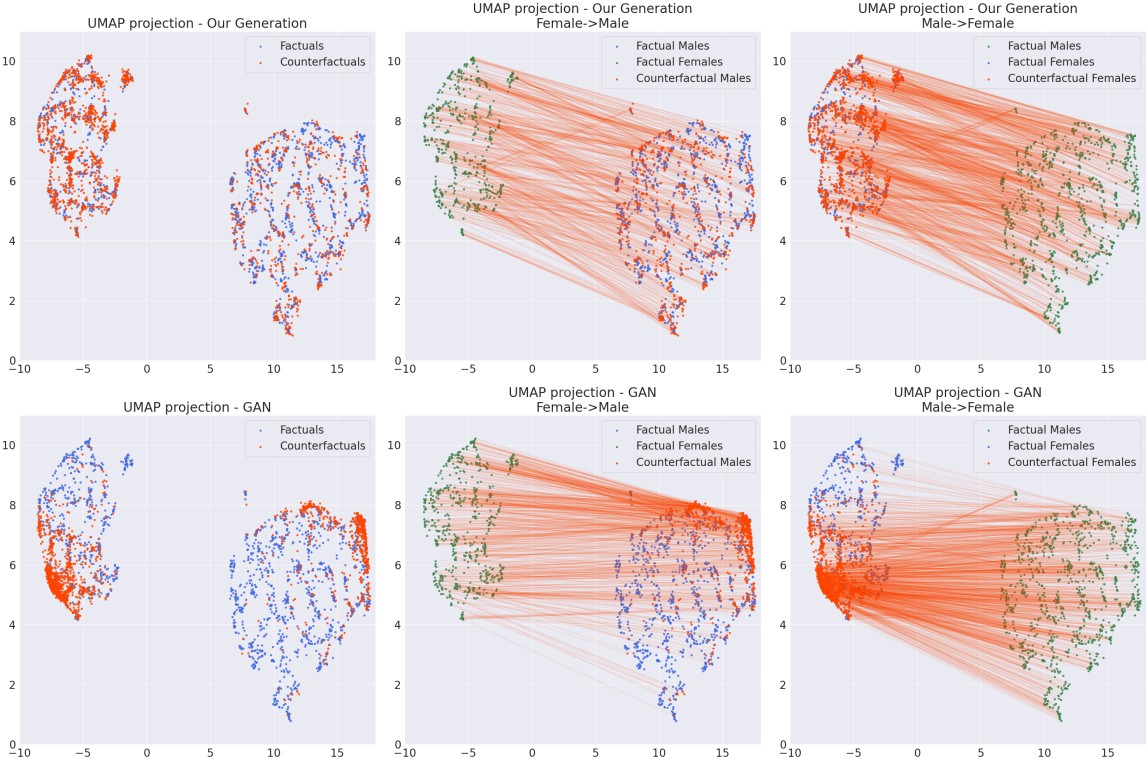

Figure 17: Samples distribution from the CardioFollow.AI dataset in a two-dimensional space after applying UMAP. Mappings for CFs generated with our method (Top Row) and through GANs (Bottom Row), showcasing all CFs(Left) or sectioned by flip: Female→ Male (Middle); Male→ Female (Right).

## Appendix F. Supplementary Examples

### F.1 Use Cases

#### F.1.1 Adult Census Income

This benchmark dataset for bias analysis in ML, provided by Kohavi and Becker (1996), is derived from the 1994 Census Bureau database and consists of 48 thousand samples. The goal is to predict whether the annual income is over or under $50k from demographic information such as age, sex, marital status, and race; and factors like education, career, and capital fluctuations. The dataset is unbalanced, with a distribution of 75.4% negative (< $50k) and 24.6% positive (> $50k) instances.

#### F.1.2 COMPAS Recidivism

The COMPAS dataset is a collection of records commonly used in the criminal justice system to predict the risk of reincidence. In our example, we employed the Adversarial Debiasing neural network model (Bellamy et al. (2018)) in two scenarios: without (base

model) and with debiasing, for the sensitive feature *Race*, which we simplified to either white or non-white individuals, for classifying if an individual is likely to reincide into crime (positive outcome, albeit negative for the individual) or not (negative outcome).

## F.2 Discussion

### F.2.1 ADULT CENSUS INCOME

To illustrate the proposed metrics' application, consider a scenario where a bank employs AI to evaluate loan applications using a model based on the Adult Census Income dataset. The system decides on loan approvals with a binary outcome: grant (positive) or deny (negative) the loan, using a minimum income threshold of $ 50,000 as a primary criterion. To ensure fairness, it is crucial to prevent any sex-based discrimination in rejecting qualified applicants. Different fairness metrics could be employed although, in this case, separation metrics would likely be preferred. To build the counterfactual world, we analysed the dataset's characteristics which comprise job titles, education level, familiar status, weekly hours, and the target variable '(annual) Income'. Through *a priori* assessment we can infer that a woman and a man with the same job and degree who input the same weekly hours and care for the same familiar core would theoretically have a similar annual income. Certain societal norms and other factors not reflected in these features may contradict this assessment; however, that in itself can be related to historical bias. Since the topic of 'Gender Wage Gaps' has been widely discussed and measures have been put in place to minimise this disparity, this model is built with the 'in world' evolution in mind. In this experiment the counterfactuals were generated by solely flipping the sensitive attribute *Sex*, believing women and men with similar professional background would have comparable incomes. Nevertheless, other strategies may be more appropriate given another context.

We trained the model using the Light GBM algorithm (Ke et al. (2017)), obtaining an ACC of 87.0% and a TPR of 66.1%. Despite the modest performance, we considered these results to suffice for demonstration purposes. Group fairness metrics yielded $9.0p.p.$ for EOpp and $7.1p.p.$ for PredEq, as detailed in Table 12. These metrics, reflecting the differences in FNR and FPR respectively, indicate a minor bias against females, evidenced by a lower FNR for males and marginally higher FPR.

**Detecting Bias**

The counterfactual metrics also show slight biases ($CMCC > 90.0\%$), although in a different perspective. We highlight a higher NSR for females (16.2% vs. 5.1%), suggesting a higher likelihood of negative outcomes when flipping to males than vice versa. We note that the FPSR for females, at 37.5%, may indicate that over a third of the FPs for this subgroup become TNs after flipping to male, hinting at possible bias in this type of error. As a loose interpretation, we could infer that the model is biased towards approving loans for females who are less likely to repay, compared to males under similar conditions.

Revisiting our bank scenario under strict EOpp legislation, with a maximum threshold of 0.05, we resorted to Fair GBM (Cruz et al. (2023)), a fairness-constrained algorithm derived from Light GBM, trained with identical hyperparameters. The resulting ECCMs are displayed in Figure 18 and the corresponding metrics are summarised in Table 12.

While EOpp improved, dropping from $9.0p.p.$ to $4.2p.p.$, our metrics showed increased counterfactual bias. The $NSR_{F \to M}$ rose from 16.2% to 25.9%, and the $FPSR_{F \to M}$ in-

Figure 18: ECCM for the models trained on the Adult Income Census data, considering the sensitive feature *Sex*. Left: Light GBM. Right: Fair GBM.
$Y$ - Ground Truth (0 for *Income* < 50k and 1 for *Income* > 50k); $G$ - Group ($M$ for *Male* and $F$ for *Female*); $\hat{Y}$ - Prediction for the factual(original) sample; $\hat{Y}_{CF}$ - Prediction for the CF sample;

Table 12: Performance and Counterfactual Metrics for the Light GBM and for the Fair GBM model trained on the Adult Income Census data, considering the sensitive feature *Sex*.

|  | Light GBM Model | | | Fair GBM Model | | |
|---|---|---|---|---|---|---|
| ACC(%) ↑ | **87.0** | 93.3 | 83.7 | 87.0 | 93.2 | 83.8 |
| TPR(%) ↑ | **66.1** | **58.6** | **67.6** | 65.4 | **61.6** | **66.2** |
| TNR(%) ↑ | **93.8** | 98.0 | 90.9 | 94.0 | 97.5 | 91.7 |
| CMCC(%) ↑ | **90.5** | 86.6 | 90.7 | 85.1 | 80.6 | 85.4 |
| PSR(%) ↓ | 2.2 | 0.7 | 3.2 | 4.4 | 0.8 | 6.8 |
| NSR(%) ↓ | 6.7 | **16.2** | 5.1 | 7.7 | **25.9** | 4.2 |
| TNSR(%) ↓ | 1.4 | 0.4 | 2.0 | 2.7 | 0.5 | 4.4 |
| FPSR(%) ↓ | 14.7 | **37.5** | 11.4 | 17.4 | **55.6** | 9.6 |
| RMSCD ↓ | 0.054 | 0.059 | 0.052 | 0.073 | 0.071 | 0.074 |
| JSCD ↓ | 0.079 | 0.130 | 0.062 | 0.097 | 0.148 | 0.083 |
|  | **Total** | $F{\rightarrow}M$ | $M{\rightarrow}F$ | **Total** | $F{\rightarrow}M$ | $M{\rightarrow}F$ |

creased from 37.5% to 55.6%, indicating that the mitigation process worsened the counterfactual bias in favour of female instances.

**Mitigating Bias - Data Augmentation with CFs**

To mitigate CF bias, we augmented the dataset with CFs altering only the sex attribute, aiming to break its typical associations. Table 13 shows that, post-augmentation, the model's performance remained stable (ACC = 87.0%, TPR = 65.8%), yet group bias increased slightly (EOpp = 12.0*p.p.*, PredEq = 7.3*p.p.*). Notably, CF bias was effectively neutralised (CR= 100.0%), as the model ceased associating the sensitive feature with others. SHAP values confirmed the sensitive feature's null contribution after augmentation, akin to its exclusion in uncorrelated scenarios. Subsequent examples involving multiple feature changes will demonstrate that this is not always the case.

**Mitigating Bias - Data Augmentation with Switched CFs**

By including only the outcome-switching CF samples, we aim to efficiently guide the model to focus on critical bias-originating samples.

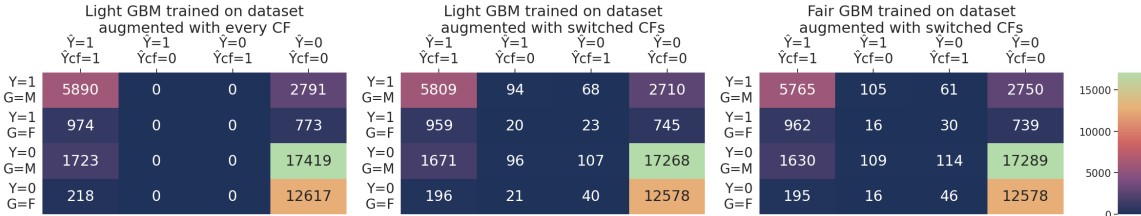

Figure 19: ECCM for the models trained on the Adult Income Census data, considering the sensitive feature *Sex*, and using data augmentation with CFs. Left: Light GBM, augmentation with all CFs. Center: Light GBM, augmented with switched CFs. Right: Fair GBM, augmented with switched CFs.
Y - Ground Truth (0 for *Income < 50k* and 1 for *Income > 50k*); G - Group (M for *Male* and F for *Female*); $\hat{Y}$ - Prediction for the factual(original) sample; $\hat{Y}_{CF}$ - Prediction for the CF sample.

The overall performance is consistent (ACC = 87.0%, TPR = 66.0%), with a minor rise in group bias (EOpp = 12.0$p.p.$, PredEq = 7.5$p.p.$). CF fairness improved significantly: overall consistency is high ($CMCC = 96.7\%$ vs. 90.5%) and the tendency to associate male candidates with lower incomes was significantly attenuated ($NSR_{F \to M} = 3.4\%$ vs. 16.2%, and $FPSR_{F \to M} = 9.7\%$ vs. 37.5%).

As a final experiment, we replaced Light GBM with Fair GBM for the augmented dataset, where the results did not display any significant improvement compared the previous instance (see Figure 19). Nonetheless, unlike the first case, it did not exacerbate CF bias, supporting that group fairness-constrained methodologies can be compatible with CF metrics (refer to Table 13). Rather, we defend group fairness and CF fairness are complementary, and efforts should be made to fulfil both criteria in model development.

### F.2.2 COMPAS RECIDIVISM

For this experiment, we will analyse the DemP, obtained from the ratio of the predicted prevalence among subgroups (%P), and employ our metrics to unveil potential biases. The standardised threshold for DemP is set at 80%, yet our initial test without debiasing revealed a rate of 0.529 (22.9% : 43.3%). This indicates that individuals who are not caucasian are twice as likely to be assigned as having a high risk of reincidence.

Other classic metrics also support this discrepancy, particularly a higher FNR and lower FPR for white individuals, resulting in an EOpp of 20.4$p.p.$ and a PredP of 14.8$p.p.$.. This suggests a bias in favour of white individuals. Our proposed metrics also report a tendency to benefit white individuals, with a slightly lower NSR (remember that the positive outcome is predicted reincidence here). Moreover, the value of 27.0% for FPSR suggests that a larger proportion of FP switch to negative (non-reincidence) when flipping other races to white.

When Adversarial Debiasing was used, it granted an increase in DemP to 0.856 (from 0.529), surpassing the legal requirement. Nevertheless, the resulting ECCM, represented in Figure 20, displayed some hidden biases derived from the mitigation process and, as a result, a tendency to impair white individuals. First, we note a higher NSR of 29.9% for white

| Augmentation | Light GBM Model | | | | | | Fair GBM Model | | |
|---|---|---|---|---|---|---|---|---|---|
| | All CFs | | | Switched CFs | | | Switched CFs | | |
| ACC (%)↑ | **87.0** | 93.2 | 83.8 | **87.0** | 93.2 | 83.7 | **87.0** | 93.3 | 83.6 |
| MCC (%)↑ | 63.4 | 64.1 | 61.2 | 63.3 | 64.3 | 60.9 | 63.3 | 64.5 | 60.8 |
| TPR (%)↑ | **65.8** | 55.8 | 67.8 | **66.0** | 56.0 | 68.0 | **65.7** | 56.0 | 67.6 |
| TNR (%)↑ | 93.9 | 98.3 | 91.0 | 93.8 | 98.3 | 90.8 | 93.9 | 98.4 | 90.9 |
| CR (%)↑ | **100.0** | **100.0** | **100.0** | 98.9 | 99.3 | 98.7 | 98.8 | 99.3 | 98.6 |
| CMCC (%)↑ | 100.0 | 100.0 | 100.0 | **96.7** | 95.3 | 96.7 | 96.4 | 95.2 | 96.5 |
| PSR (%)↓ | 0.0 | 0.0 | 0.0 | 0.7 | 0.5 | 0.9 | 0.7 | 0.6 | 0.9 |
| NSR (%)↓ | 0.0 | 0.0 | 0.0 | 2.6 | **3.4** | 2.5 | 2.8 | **2.7** | 2.8 |
| TNSR (%)↓ | 0.0 | 0.0 | 0.0 | 0.5 | 0.3 | 0.6 | 0.5 | 0.4 | 0.7 |
| FPSR (%)↓ | 0.0 | 0.0 | 0.0 | 5.9 | **9.7** | 5.4 | 6.4 | **7.6** | 6.3 |
| RMSCD↓ | 0.00 | 0.00 | 0.00 | 0.030 | 0.036 | 0.026 | 0.027 | 0.032 | 0.025 |
| JSCD↓ | 0.00 | 0.00 | 0.00 | 0.053 | 0.092 | 0.038 | 0.049 | 0.082 | 0.035 |
| | **Total** | $F_{\rightarrow}M$ | $M_{\rightarrow}F$ | **Total** | $F_{\rightarrow}M$ | $M_{\rightarrow}F$ | **Total** | $F_{\rightarrow}M$ | $M_{\rightarrow}F$ |

Table 13: Performance and Counterfactual Metrics for the Light GBM and for the Fair GBM model trained on the Adult Income Census data augmented with all or only the switched CFs, considering the sensitive feature *Sex*.

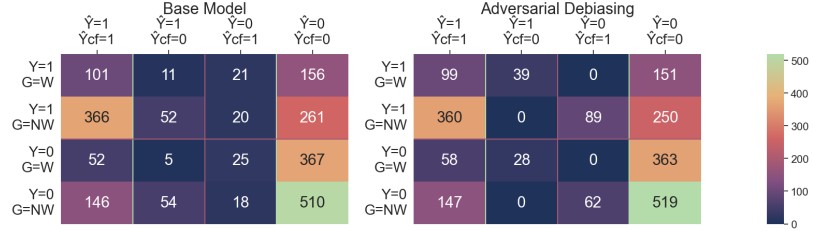

Figure 20: The ECCM generated for the base model and for the Adversarial Debiasing model trained on the COMPAS data, considering the sensitive feature *Race*. $Y$ - Ground Truth (0 for *Low risk of criminal recidivism* and 1 for *High risk of criminal recidivism*); $G$ - Group ($W$ for *White* and $NW$ for *Not White*); $\hat{Y}$ - Prediction for the factual(original) sample; $\hat{Y}_{CF}$ - Prediction for the CF sample.

instances, compared to 0.0% for other races. Additionally, inspecting the metrics including the ground truth, we observe a higher likelihood for white samples to switch TP to FN (28.3% vs 0.0%), and FP to TN (32.6% vs 0.0%) when flipping *Race*. On the other hand, when switching the sensitive feature to white, there is a propensity to detect previously overlooked cases (FN) for non-whites, noted by a FNSR of 26.3%. These findings highlight the need for complementary evaluation frameworks for fairness in ML since optimising towards specific criteria may introduce other types of undesirable biases.

Table 14: Classic and Counterfactual metrics obtained for the COMPAS dataset before and after applying fairness constraints.

|  | Base Model | | | Adversarial Debiasing | | |
|---|---|---|---|---|---|---|
| FNR (%) ↓ | 46.4 | **61.2** | **40.2** | 49.6 | 52.2 | 48.5 |
| FPR (%) ↓ | 21.8 | **12.7** | **27.5** | 19.8 | 19.2 | 20.2 |
| %P (%) | 36.4 | **22.9** | **43.3** | 33.8 | **30.4** | **35.5** |
| CMCC (%) ↑ | 79.3 | 78.1 | 79.6 | 78.4 | 78.7 | 80.3 |
| SR (%) ↓ | 9.5 | 8.4 | 10.1 | 10.1 | 9.1 | 10.6 |
| PSR (%) ↓ | 6.1 | 8.1 | 4.7 | 10.5 | 0.0 | 16.4 |
| NSR (%) ↓ | 15.5 | **9.5** | **17.2** | 9.2 | **29.9** | 0.0 |
| TPSR (%) ↓ | 11.9 | 9.8 | 12.4 | 7.8 | 28.3 | 0.0 |
| FPSR (%) ↓ | 23.0 | 8.8 | **27.0** | 12.0 | 32.6 | 0.0 |
| TNSR (%) ↓ | 4.7 | 6.4 | 3.4 | 6.6 | 0.0 | 10.7 |
| FNSR (%) ↓ | 9.0 | 11.9 | 7.1 | 18.2 | 0.0 | **26.3** |
| RMSCD ↓ | 0.046 | 0.047 | 0.045 | 0.077 | 0.078 | 0.075 |
| JSCD ↓ | 0.037 | 0.036 | 0.039 | 0.062 | 0.039 | 0.042 |
|  | Total | W→NW | NW→W | Total | W→NW | NW→W |

### F.2.3 Heart Disease

In this experiment, bias associated with *Sex* is evaluated. Concerning the correlation of the features with the sex of the individual, there are subtle differences between electrocardiograms for men and women due to hormonal levels, especially the impact of estrogen and testosterone in cardiac functions, as well as anatomical differences such as the size of the heart (Knowlton and Lee (2012)). As an example, women tend to have higher resting heart rates due to having on average smaller hearts, needing higher frequency to pump enough blood. Additionally, the protective effect of estrogen in arteries is heavily documented, explaining the increased risk of cardiac complications after menopause for women (Bokhari and Bergmann (2002)).

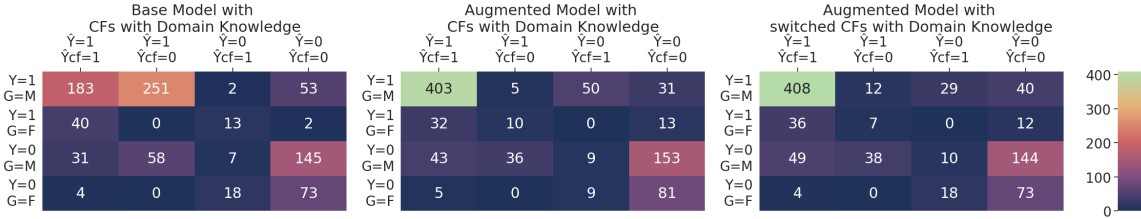

Figure 21: ECCM for the base model trained on the Heart Disease dataset, considering the sensitive feature *Sex*, and using data augmentation with CFs generated with Domain Knowledge. Left: Base Model; Middle: Model augmented with all the CFs; Right: Model augmented with switched CFs.
*Y* - Ground Truth (0 for *Free from CAD* and 1 for *Signs of CAD*); *G* - Group (*M* for *Male* and *F* for *Female*); $\hat{Y}$ - Prediction for the factual(original) sample; $\hat{Y}_{CF}$ - Prediction for the CF sample.

**Analysing Bias - CFs with Domain Knowledge**

This dataset contains several attributes extracted from the Eletrocardiogram (ECG), which were taken into account for generating CFs. In contrast, features such as *Age* and pre-existing conditions and risk factors were kept unchanged.

The base model is a Logistic Regression and despite the small sample size, the results, shown in Table 15, are satisfactory with an $ACC = 81.5\%$ and $TPR = 87.1\%$. Nonetheless, there is considerable bias noted by a difference in TPR against women, $PredP = 16.1p.p.$. This discrepancy is apparent in the majority of the cross-validation scheme, reinforcing the existence of model bias. Drawing the ECCM (c.f. Figure 21), the results display a clear tendency to associate female patients with negative outcomes, $TPSR_{M \to F} = 57.8\%$ and $NSR_{M \to F} = 59.1\%$, as well as a tendency of associating positive outcomes to men, $TNSR_{F \to M} = 19.8\%$ and $PSR_{F \to M} = 29.2\%$, supporting the group fairness assessment.

**Mitigating Bias - Augmentation with all the CFs**

To mitigate this bias, the dataset was supplemented with all the CFs generated in the first instance. Using this augmented set, the CF bias was mostly mitigated, but, unexpectedly, it introduced an opposite effect leading $TP_F$ to switch to $FN_M$, as per a $TPSR_{F \to M} = 23.8\%$. While high, considering the $TPSR_{M \to F}$ for the base model was substantially higher and the other mentioned metrics decreased, the augmentation can be deemed successful in the mitigation of CF bias.

Moreover, since the performance decreased, there is room to explore other iterations that hopefully achieve similar results without duplicating the sample space. A logical conclusion is to select which CFs to include. There are different available routes to achieve this, such as selecting the most unique CFs to avoid redundancy, or selecting CFs that support the Domain Knowledge, stirring the model in a predetermined direction as it is displayed in experience Section 3.1. Finally, there is a simple approach of including CFs that show a different outcome than the original sample, called 'switched CFs'.

**Mitigating Bias - Augmentation with switched CFs**

Since the model is already able to correctly predict a given set of pairs of samples and their CFs, there seems to be no need to add information about the CF equivalent of these samples. Thus, only the incorrectly predicted, or switched CFs, should be added. This method is not infallible, due to the unpredictability of training a model, but it can achieve good results in specific scenarios. Compared to the original model, it was successful, in this case, in mitigating CF, the CMCC increased from 33.1% in the original model to 72.2%, and group bias, as per a $PredP = 7.7p.p.$, with minimal loss in total performance, as seen by an $ACC = 80.5\%$ and $TPR = 85.1\%$. In relation to augmentation with all the CFs the introduced bias displayed that severely increased the $TPSR_{F \to M}$, is much more tenuous, $TPSR_{F \to M} = 16.3\%$. In contrast, it was not able to correct the flux of $TN_F$ to $FP_M$, maintaining the $TNSR_{M \to F}$ value at 19.8%. Overall, given this classification context application that poses recall as the priority, opting to add only switched CFs proved to be more fruitful in mitigating bias in the priority class. Along with the fact that the contested metric in this case, $TNSR_{M \to F}$, is largely less grave in value than the $TPSR_{F \to M}$ introduced by augmentation with all the CFs.

Table 15: Performance and Counterfactual metrics for the base model and models trained on the dataset augmented with all the CFs and only the CFs that switched.

| | | | | Augmentation Method | | | | | |
| | Base Model | | | All CFs | | | Switched CFs | | |
|---|---|---|---|---|---|---|---|---|---|
| ACC (%) ↑ | **81.5** | 87.3 | 80.3 | 79.8 | 88.0 | 78.1 | **80.5** | 89.3 | 78.6 |
| MCC (%) ↑ | 60.3 | 72.5 | 54.1 | 57.4 | 73.9 | 50.6 | 58.4 | 76.8 | 50.8 |
| TPR (%) ↑ | **87.1** | **72.7** | **88.8** | 82.7 | **76.4** | **83.4** | **85.1** | 78.2 | **85.9** |
| TNR (%) ↑ | 72.3 | 95.8 | 63.1 | 75.0 | 94.7 | 67.2 | 72.9 | 95.8 | 63.9 |
| CMCC (%) ↑ | **33.1** | 64.5 | 35.8 | 71.5 | 70.4 | 68.7 | **72.2** | 64.4 | 71.7 |
| SR (%) ↓ | 39.7 | 20.7 | 43.6 | 13.5 | 12.7 | 13.7 | 13.0 | 16.7 | 12.2 |
| PSR (%) ↓ | 12.8 | **29.2** | 4.4 | 19.7 | 8.7 | 24.3 | 17.5 | 17.5 | 17.5 |
| NSR (%) ↓ | 54.5 | 0.0 | **59.1** | 9.6 | 21.3 | 8.4 | 10.3 | 14.9 | 9.9 |
| TPSR (%) ↓ | 53.0 | 0.0 | **57.8** | 3.3 | **23.8** | 1.2 | 4.1 | **16.3** | 2.9 |
| FNSR (%) ↓ | 33.1 | 64.4 | 35.8 | 71.5 | 70.4 | 68.7 | 72.2 | 64.4 | 71.7 |
| TNSR (%) ↓ | 10.3 | **19.8** | 4.6 | 7.1 | 10.0 | 5.6 | 11.4 | **19.8** | 6.5 |
| FPSR (%) ↓ | 62.4 | 0.0 | 65.2 | 42.9 | 0.0 | 45.6 | 41.8 | 0.0 | 43.7 |
| RMSCD ↓ | 0.368 | 0.263 | 0.386 | 0.172 | 0.159 | 0.175 | 0.146 | 0.138 | 0.147 |
| JSCD ↓ | 0.243 | 0.125 | 0.211 | 0.120 | 0.163 | 0.113 | 0.101 | 0.121 | 0.096 |
| | **Total** | $F{\rightarrow}M$ | $M{\rightarrow}F$ | **Total** | $F{\rightarrow}M$ | $M{\rightarrow}F$ | **Total** | $F{\rightarrow}M$ | $M{\rightarrow}F$ |

### F.2.4 CardioFollow.AI - Bias for the sensitive feature *Sex*

The proposed method for generating CFs (c.f. Section 2.1), ensures minimum changes in the original sample. However, it mostly replicates the already preexisting patterns in the original dataset. In this section, two sets of generated CFs are analysed: one in which all the features have the potential to change, and another with only feasible changes, based on Domain Knowledge. In this experience, instead of the deployed model described in Section 2.5.1, a model following the same architecture but trained with all the extracted features is used. This is done to include features directly related to the sensitive feature, such as 'BMI', 'Body Surface Area (BSA)', 'Weight', 'Height', and 'Creatinine'. If not included, the generated CFs incur the risk of not suffering any alterations and, for this reason, there are no possible interpretations to draw from. The training set is built from the contributions of 1557 women and 2416 men with approximately the same prevalence, approximately 7%. In addition to comparing different CFs sets for bias analysis, different augmentation approaches are studied. The different metrics for the resulting models are summarised in Table 16 to allow an easier discussion. Each metric is extrapolated from the sum of the five matrixes correspondent to each fold generated from cross-validation. These sets are validated through real-world knowledge.

**Analysing Bias - CFs without Domain Knowledge**

Before delving into CFs, the first step is to evaluate the base model performance and group fairness metrics. In this specific task, where the TPR is optimised, the most suited group fairness metric is PredP. The base model has an overall TPR of 74.6%. Sectioning by the sensitive feature, the male subgroup has 72.3% TPR and the female subgroup has 78.0% TPR. In terms of PredP this corresponds to a 5.7*p.p.* difference, indicating slight bias against men. As for the negative outputs, the overall TNR is 48.8%, meaning the model predicts just as much FP as TN. This statistic is relatively consistent in both sexes with 49.4% for males and 47.8% for females.

Figure 22: ECCM for the base model with CFs generated with (right) and without (left) domain knowledge.
$Y$ - Ground Truth (0 for *Uneventful Recoveries* and 1 for 'post-surgery complications'); $G$ - Group ($M$ for *Male* and $F$ for *Female*); $\hat{Y}$ - Prediction for the factual(original) sample; $\hat{Y}_{CF}$ - Prediction for the CF sample.

Evaluating the counterfactual fairness of the model with CFs generated without Domain Knowledge, there are more prediction changes when flipping male to female than the other way around, $SR_{M \rightarrow F} = 11.7\%$ and $SR_{F \rightarrow M} = 8.5\%$, and there is better consistency regardless of the outcome as well, as seen by $CMCC_{F \rightarrow M} = 83.2\%$ against $CMCC_{M \rightarrow F} = 77.2\%$. By itself, this is indicative of a more considerable bias in female patients, but other metrics support a tendency to associate female patients with positive outcomes, such as $PSR_{M \rightarrow F} = 19.7\%$. A large portion of these positive flips are correct predictions, $FNSR_{M \rightarrow F} = 36.7\%$, suggesting that more than one-third of the FNs predicted in male samples were correctly predicted as TPs, once changed to CF females. This seems to be counter-intuitive as it would be expected that changing from men to women would lead to less FNs, to match the original female TPR. Nonetheless, this result can unveil bias that would only be revealed with new predictions on samples less similar to the training dataset. However, it is important to note how these CFs were generated. Because they were created without considering Domain Knowledge, the modifications incurred tend to display the typical sample of the other subgroup. For example, one heavily correlated feature with sex is *Smoking*, a risk factor much more common in men than women in this dataset. 47.0% of men and only 9.9% of women are smokers, meaning 88.0% of smokers are men. Based on these statistics, when generating a CF for a male smoker, the result is most likely a woman who does not smoke. Logically, *Smoking* should be unrelated to the sex of the patient and, as an addictive trait and risk factor, it should be kept unchanged in the CF.

**Analysing Bias - CFs with Domain Knowledge**

Considering this setback, by resorting to Domain Knowledge, the second set of CF was generated only allowing changes in the features that should be related to the patient's sex. The existing preconditions and risk factors should be left untouched since they are the most critical points of similarity in this task. The main features that can be justifiably switched are related to general biological differences between men and women. Men tend to have lower fat storage, and higher density bones and are generally taller than women (Schlecht et al. (2015); Power and Schulkin (2008)). Men tend to have higher muscle mass, resulting in slightly higher creatinine levels than women. Creatinine is a waste product produced by

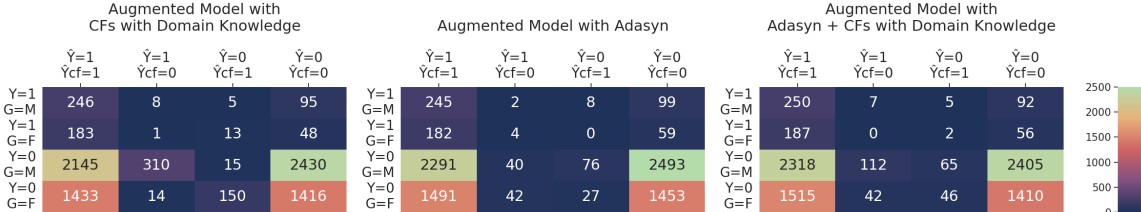

Figure 23: ECCM for base model and the model after data augmentation with CFs, ADASYN, and a combination of both.
$Y$ - Ground Truth (0 for *Uneventful Recoveries* and 1 for *Post-surgery Complications*); $G$ - Group ($M$ for *Male* and $F$ for *Female*); $\hat{Y}$ - Prediction for the factual(original) sample; $\hat{Y}_{CF}$ - Prediction for the CF sample.

muscle metabolism and excreted by the kidneys and high levels can indicate kidney disfunction. Thus, for the same blood concentration, women may face a more severe condition than men (Agrawal et al. (2015)). The potential features for generating the CFs are 'BMI', 'BSA', 'Weight', 'Height', and 'Creatinine'.

Contrary to the tendency displayed with CFs generated without Domain Knowledge, this set reveals a tendency to associate male samples with positive outcomes, given by $PSR_{F \to M} = 8.1\%$, with a significant portion being from switches from FNs to TPs, $FNSR_{F \to M} = 18.5\%$. At the same time, it is noted a tendency to associate negative outcomes to female samples, $NSR_{M \to F} = 7.4\%$. This is the exact opposite of the other set of CFs, emphasising the importance of a careful CF generation process. Assuming these CFs as plausible, these results point to a slight bias against women. To attempt to mitigate this issue, the dataset was augmented with the CFs generated for the train set in an attempt to teach the model to better understand the differences in how to predict female and male samples.

The ECCM for CFs with and without Domain Knowledge can be visualised in Figure 22, where one can visually observe less samples in the middle columns when Domain Knowledge is used, suggesting lower counterfactual bias.

**Augmentation with CFs**

Augmenting the dataset with CFs has the purpose of teaching the model how it should interpret each instance of symptoms and characteristics in the case of the patient being male or female.

However, applying the method to the training set did not achieve better results in the criteria of performance, slightly decreasing the TPR at 73.1% and only increasing TNR by $1.9p.p.$ to 50.7%. It was able to mitigate some of the discrepancies between men and women in TPR, $TPR_M = 71.8\%$ and $TPR_F = 75.1\%$, resulting in a $PredP = 3.3p.p..$ However, the difference is so subtle, it poses doubt about its actual improvement, especially considering the drawbacks of data augmentation. Analysing the CFs changes, the method is proved as not effective in this particular instance. The model displays a higher tendency to associate male samples with positive outcomes $PSR_{F \to M} = 10.0\%$, with a larger portion occurring from TNs to FPs, $TNSR_{F \to M} = 9.6\%$. On the other hand, the tendency to

associate female samples with negative outcomes seems to be more prominent as well, $NSR_{M \to F} = 11.7\%$, with a higher portion resulting from TPs to FNs, $TPSR_{M \to F} = 3.2\%$. This is an unexpected result as it would be expected that augmentation with CFs would decrease CF bias. Although the metrics' values do not infer substantial bias, this experience reveals that, based on context, including all the CFs may not be beneficial.

**Augmentation with Adaptive Synthetic Sampling (ADASYN)**

It is established the potential problem of this task lies in insufficient samples, hinting at augmentation as a viable option. There are several techniques for data augmentation, and the ADASYN method is well-regarded for its ability to oversample less populated distributions by using density-based sample saturation to generate additional synthetic samples (He et al. (2008)).

Proceeding with this method, the overall performance remained similar, but the group performance became more discrepant in terms of TPR, $TPR_M = 69.8\%$ and $TPR_F = 75.9\%$, $PredP = 6.2p.p.$ and TNR, $TNR_M = 52.4\%$ and $TNR_F = 49.1\%$. Nevertheless, there are fewer switches in the CF predictions, $CMCC = 95.3\%$. The $NSR_{M \to F} = 1.6\%$ and the $TPSR_{M \to F} = 0.8\%$ are considerably lower. This method led to a worse equilibrium between male and female samples in terms of group fairness. It is important, however, to retain the properties that allowed for better CFs parameters.

**Mitigating Bias - Augmentation with CFs oversampling with ADASYN**

Trying to improve the overall performance, the dataset was augmented with CFs and then oversampled with ADASYN. CFs serves to increase variability in the data, for example, with more female patients that smoke, while the oversampling technique allows to fill less populated feature spaces. As a result, the performance slightly improved in relation to the other augmentation methods, $TPR = 74.1\%$. Analysing by group, the TPR improved slightly for men and decreased for women, $TPR_F = 76.3\%$ and $TPR_M = 72.6\%$, with $PredP = 3.6p.p.$. For negative outcomes, the values improve in relation to the original model but are worse than the other tested techniques. As for the CF analysis, there are fewer flips, with the overall CMCC improving from 89.7% to 93.4%, mitigating all the mentioned metrics in the base model by at least $3.0p.p.$.

This use case displays how the generation of CFs heavily influences the results and the importance of integrating Domain Knowledge to ensure a reliable evaluation. As for mitigation exploration, this experience allowed to delve into the implications posed by CF augmentation, as well as the potential for combining these samples with other methods to allow for a more robust model in terms of performance and bias.

The ECCM for each of the augmentation techniques can be visualised in Figure 23, complemented by the most relevant metrics summarised in Table 16.

F.2.5 CARDIOFOLLOW.AI - BIAS FOR THE SENSITIVE FEATURE *Smoking*

Suspecting the potential bias due to the lack of representation of specific samples, the next step is to attempt mitigation through augmentation with CFs. From this, it is expected to mitigate both group and CF bias. Retraining the model with the dataset augmented by the train set CFs, the results show improvement in performance for smokers, $TPR_S = 72.8\%$, and non smokers, $TPR_S = 77.0\%$, both accompanied by a slight decrease in the negative outcomes, where total $TNR$ decreases from 50.2% to 49.1%.

Table 16: Performance and Counterfactual Metrics referring to the base model and the model after data augmentation with ADASYN, plausible CFs and a combination of both.

| | Augmentation Method | | | | | | | | | | | |
| | Base | | | Plausible CFs | | | ADASYN | | | CFs with ADASYN | | |
|---|---|---|---|---|---|---|---|---|---|---|---|---|
| MCC(%)↑ | 12.0 | 13.6 | 10.9 | 12.2 | 14.3 | 10.9 | 12.0 | 13.2 | 11.1 | 12.2 | 13.0 | 11.5 |
| TPR(%)↑ | **74.6** | **78.0** | **72.3** | **73.1** | **75.1** | **71.8** | **72.3** | **75.9** | **69.8** | **74.1** | **76.3** | **72.6** |
| TNR(%)↑ | **48.8** | **47.8** | **49.4** | 50.7 | 52.0 | 49.9 | 51.2 | **49.1** | **52.4** | 49.6 | 48.3 | 50.4 |
| CMCC(%)↑ | **89.7** | 89.9 | 89.8 | 87.9 | 89.4 | 87.7 | **95.3** | 95.5 | 95.2 | **93.4** | 94.4 | 92.8 |
| SR(%)↑ | 5.1 | 5.1 | 5.2 | 6.1 | 5.5 | 6.4 | 2.3 | 2.2 | 2.4 | 3.3 | 2.8 | 3.6 |
| PSR(%)↓ | 4.6 | **8.1** | 2.8 | 4.4 | **10.0** | 0.8 | 2.6 | 1.8 | 3.1 | 2.9 | 3.2 | 2.7 |
| NSR (%)↓ | 5.5 | 2.5 | **7.4** | 7.8 | 0.9 | **11.7** | 2.0 | 2.7 | **1.6** | 3.6 | 2.4 | 4.4 |
| TPSR(%)↓ | 1.8 | 1.1 | 2.3 | 2.1 | 0.5 | **3.2** | 1.4 | 2.2 | **0.8** | 1.6 | 0.0 | 2.7 |
| TNSR(%)↓ | 4.5 | 7.7 | 2.6 | 4.1 | **9.6** | 0.6 | 2.5 | 1.8 | 3.0 | 2.8 | 3.2 | 2.6 |
| FPSR(%)↓ | 5.9 | 2.7 | 7.9 | 8.3 | 1.0 | 12.6 | 2.1 | 2.7 | 1.7 | 3.9 | 2.7 | 4.6 |
| FNSR(%)↓ | 10.5 | **18.5** | 6.1 | 11.2 | 21.3 | 5.0 | 4.8 | 0.0 | 7.5 | 4.5 | 3.5 | 5.2 |
| RMSCD | 0.014 | 0.014 | 0.014 | 0.019 | 0.018 | 0.020 | 0.011 | 0.011 | 0.010 | 0.013 | 0.013 | 0.014 |
| JSCD | 0.012 | 0.012 | 0.012 | 0.016 | 0.013 | 0.014 | 0.009 | 0.009 | 0.008 | 0.012 | 0.011 | 0.0121 |
| | **Total** | $F{\rightarrow}M$ | $M{\rightarrow}F$ | **Total** | $F{\rightarrow}M$ | $M{\rightarrow}F$ | **Total** | $F{\rightarrow}M$ | $M{\rightarrow}F$ | **Total** | $F{\rightarrow}M$ | $M{\rightarrow}F$ |

When reevaluating the CF metrics, it was noted that the value of $TNSR_{S \rightarrow NS} = 12.1\%$ was mitigated (vs. 54.5%), as well as $TPSR_{NS \rightarrow S} = 0.3\%$ (vs. 10.9%). Nevertheless, the value of $FPSR_{NS \rightarrow S} = 57.4\%$ is significantly higher than the base model without augmentation (12.6%). When there is augmentation, a high flux from initially incorrectly predicted values to correctly predicted values may indicate some overfitting. This occurs because the CFs for the test group are generated based on the train samples, thus they are inevitably closer to the train set than the original samples. For this reason, the main concern is the switches in original TPs and TNs.

