# OpenReview forum: "The Matrix Reloaded: Towards Counterfactual Group Fairness in Machine Learning"
_DMLR — Accepted by DMLR_

### Review · Reviewer_wLEM · 2024-03-04

**Recommendation:** 3
**Confidence:** 2

**Summary Of Contributions:**

This paper introduces a novel framework for evaluating bias in machine learning models through the generation of plausible counterfactuals and a Counterfactual Confusion Matrix (CCM). This approach allows for a more nuanced analysis of bias by considering changes in sensitive attributes and their impact on model predictions. The methodology is flexible, incorporating domain knowledge into generating counterfactuals, and provides a suite of metrics derived from the CCM for a comprehensive bias analysis. The framework is demonstrated on real-world datasets, showing its capability to uncover subtle biases that traditional methods might miss.

**Strengths:**

Please see the above section **Strengths And Weaknesses**.

**Audience:**

Yes

**Claims And Evidence:**

The claims regarding the framework's ability to reveal subtle biases and improve fairness in machine learning models are well supported by experiments on real-world datasets. The paper thoroughly analyzes the results, comparing them to traditional bias evaluation methods and demonstrating the added value of the proposed counterfactual metrics.

**Datasets And Benchmarks:**

N/A

**Extended Submissions:**

N/A

**Limitations:**

Please see the above section **Strengths And Weaknesses**.

**Requested Changes:**

(1) **Grammer issues:** i.e., at the end of page 2, "enrolment". Besides, it would be better to modify the usage of quotation marks.

(2) **Clarity:** It would be much better if authors could repeat the full name of the abbreviation frequently, i.e., in the caption of Figure 3, the author could use "Figure 3: The Extended Counterfactual Confusion Matrix (ECCM)" or more detailed descriptions, instead of  "Figure 3: ECCM". Besides, many figure captions appear to be too brief, which is not very easy for readers to understand.

(3) **Notations:** Several notations are used without explanations, or may cause confusion, to name a few, $X_{A_+}$ in the line before Eqn. (1) and $v_j$ in Eqn. (4) are never clearly defined, the usage of $v_{i+1}$ and $v_{i-1}$ are kind of confusing.

(4) **Illustration of experiment results:** In tables of experiment results, i.e., in Table 2, what is the meaning of statistics in bold? It would be much better if the authors could explain the experiment results/numbers by clearly stating how they demonstrate the effectiveness of the proposed metrics.

(5) No section name for Appendix C.

**Strengths And Weaknesses:**

**Strengths**

* **Innovative Bias Evaluation Framework:** The authors introduce a flexible approach to generate plausible counterfactuals using probabilistic distributions and domain knowledge, enabling nuanced bias analysis beyond traditional methods.

* **Counterfactual Confusion Matrix (CCM):** The authors propose the CCM and derived metrics for comparing model outcomes in original and counterfactual scenarios, providing insights into the model's fairness and resilience to changes in sensitive attributes.

**Weaknesses**

* **The writing of the paper could be further improved:** Please refer to the section **Request Changes** for more details (i.e., grammar issues, clarity of figure captions, math notations, experiment results, etc).

* **Dependency on Domain Knowledge:** The effectiveness of the counterfactual generation heavily relies on domain knowledge, which might not always be readily available or accurately codified.

* **Scalability and Generalization (minor):** The counterfactual generation process might not scale well to very large or complex datasets, and its applicability to non-binary or multi-class scenarios is not well discussed.

---

### Review · Reviewer_o1uo · 2024-03-11

**Recommendation:** 3
**Confidence:** 2

**Summary Of Contributions:**

The paper proposed a comprehensive framework for evaluating and mitigating bias in machine learning through counterfactual reasoning.

**Strengths:**

1. The paper validates the proposed approach using real-world datasets. The use of real-world datasets connects the theoretical framework to real-world applications, which is a crucial step in demonstrating the applicability and effectiveness of the framework.
2.  The paper introduces a Counterfactual Confusion Matrix and derived metrics, offering unique insights into model behavior under counterfactual conditions. This expands on the previous fairness and interoperability research by providing a more nuanced analysis of the model behavior. This helps the broader research community to improve model fairness.
3. The paper addresses both the identification and mitigation of biases, e.g., using data augmentation with counterfactual examples. This helps the community develop fair machine learning models.
4. The paper is well-written and easy to follow.

**Audience:**

Yes

**Broader Impact Concerns:**

No broader impact concerns.

**Claims And Evidence:**

The claims made in the submission are supported by accurate, convincing, and clear evidence. The authors conducted empirical validation on the real-world dataset to validate the proposed framework. The authors also analyzed the model's performance using a comprehensive set of bias metrics.

**Datasets And Benchmarks:**

The paper uses 1) the Adult Census Income Dataset (derived from the 1994 Census Bureau database) and 2) the CardioFollow.AI Dataset. The dataset sources are properly cited, and the paper provides a detailed description of the dataset. No ethical concerns.

**Extended Submissions:**

The authors did not mention the paper is an extended submission.

**Limitations:**

1. The authors reference existing methods in counterfactual reasoning and fairness in machine learning (e.g., counterfactual explanations, the distinction between counterfactual features v.s. outcomes, counterfactual bias evaluation,  counterfactual bias mitigation). However, no direct and quantitative comparisons are offered for existing approaches.
2. The framework relies on domain knowledge for generating counterfactual scenarios. The authors outline a process for generating counterfactuals that adjust not just the sensitive feature (e.g., sex or race) but also other features correlated with the sensitive feature. This process is informed by domain knowledge, which helps identify which features should be adjusted and how to create realistic scenarios where the only significant changes are those that directly relate to the counterfactual hypothesis. Here, domain knowledge plays a crucial role in determining how features related to the sensitive attribute are adjusted. However, the usage of the framework will be limited in cases where a universally accepted counterfactual scenario cannot be generated.
3. The paper did not perform a detailed analysis of the trade-off between mitigating bias and the model performance. For example, the impact on accuracy, precision, and recall.

**Requested Changes:**

1. Compare the proposed method to existing methods in counterfactual reasoning in detail.
2. Discuss the scenario where a universally accepted counterfactual scenario cannot be generated, and how to adjust the framework.
3. Analyze the trade-off between mitigating bias and model performance.

See limitations below for details.

**Strengths And Weaknesses:**

Strength:
1. The paper evaluated the framework on real-world datasets.
2. The paper introduces a Counterfactual Confusion Matrix and derived metrics.
3. The paper discusses how to identify biases and also how to mitigate them.

See strengths below for details.

Weakness:
1. The paper lacks direct, quantitative comparisons with these existing approaches.
2.  Domain knowledge is essential in the paper's framework for generating plausible counterfactual scenarios. However, the usage of the framework may be limited where no consensus on counterfactual scenarios exists.
3. Lack of detailed analysis of the trade-off between bias mitigation and model performance

See limitations below for details.

---

### Review · Reviewer_o9dt · 2024-03-31

**Recommendation:** 2
**Confidence:** 2

**Summary Of Contributions:**

The paper proposes extended counterfactual confusion matrix to describe model's behavior under counterfactual conditions. Specifically, the paper combines counterfactual explanation with group-level summary statistics (e.g., in the format of a confusion matrix) and evaluates the model via generated counterfactual examples. The empirical evaluations are presented on both publicly-available and proprietary data sets.

---

I confirm that I have carefully read authors' response, as well as comments from other reviewers.

**Strengths:**

Overall the paper is not hard to follow. The motivation and the proposed approach are well-presented.

**Audience:**

Yes

**Broader Impact Concerns:**

I do not have any concerns on the ethical implications of the work that would require adding a Broader Impact Statement.

**Claims And Evidence:**

The claim that "[the ECCM approach provides] unique insights into the model’s resilience and susceptibility to changes in sensitive attributes" may need further clarifications and discussions.

**Datasets And Benchmarks:**

The paper utilizes two data sets: (1) the publicly available Adult data set, and (2) the proprietary CardioFollow.AI data set. The paper contains descriptions on the data sets in terms of features utilized.

**Extended Submissions:**

N/A

**Limitations:**

The ECCM approach provides additional ways to compare group-level summary statistics, but it remains more or less unclear (in the sense of the lack of operationable guidance) how one should interpret such confusion matrix to enhance the model, especially when there are multiple potential conclusions one can draw from ECCM.

**Requested Changes:**

It would be better if the takeaway message can be made clearer, w.r.t. the "unique insights into the model’s resilience and susceptibility to changes in sensitive attributes." In particular, what are the intended insights about the model that the paper would like to reveal? Additional points can be further clarified in order to answer this question:

- If model's resilience to changes in sensitive attributes (especially when taking ECCM into consideration) is the target, does it mean that we should actually go beyond group-level fairness to the individual-level Counterfactual Fairness (Kusner et al. 2017)?

- In ECCM, there are multiple different comparisons one can draw, which one(s) should we use, and why? While the paper mentions the potential need of including of domain knowledge, are there any takeaway messages or operationable guidelines for practitioners that can specify how one should interpret and utilize ECCM?

- By investigating ECCM, how one can improve the model? Do conclusions draw from difference comparisons on ECCM entries yield a single direction for model improvement? If not, how should we proceed?

Minor thing: all left quotation marks need to be fixed

**Strengths And Weaknesses:**

Overall the paper is not hard to follow. The motivation and the proposed approach are well-presented.

While it is nice to see the summary of group-level summary statistics (based on counterfactual explanations/reasoning) into the format of an extended confusion matrix, the technical approach itself is relatively intuitive, especially considering the existing characterization (e.g., Coston et al., 2020) of counterfactual group-level fairness in previous works. The "extension" of counterfactual confusion matrix is more or less direct, in the sense that there is an additional axis of consideration: to compare CF prediction and original prediction. The approach's intended "unique insights into the model’s resilience and susceptibility to changes in sensitive attributes" are still a little bit confusing (question detailed in "Requested Changes" below).